# Enhancing Motion Deblurring in High-Speed Scenes with Spike Streams

**Shiyan Chen**[1,2*]    **Jiyuan Zhang**[1,2*]    **Yajing Zheng**[1,2†]    **Tiejun Huang**[1,2,3]    **Zhaofei Yu**[1,2,3†]

[1]School of Computer Science, Peking University
[2]National Key Laboratory for Multimedia Information Processing, Peking University
[3]Institute for Artificial Intelligence, Peking University
`{strerichia002p,jyzhang}@stu.pku.edu.cn,`
`{yj.zheng,yuzf12,tjhuang}@pku.edu.cn`

## Abstract

Traditional cameras produce desirable vision results but struggle with motion blur in high-speed scenes due to long exposure windows. Existing frame-based deblurring algorithms face challenges in extracting useful motion cues from severely blurred images. Recently, an emerging bio-inspired vision sensor known as the spike camera has achieved an extremely high frame rate while preserving rich spatial details, owing to its novel sampling mechanism. However, typical binary spike streams are relatively low-resolution, degraded image signals devoid of color information, making them unfriendly to human vision. In this paper, we propose a novel approach that integrates the two modalities from two branches, leveraging spike streams as auxiliary visual cues for guiding deblurring in high-speed motion scenes. We propose the first spike-based motion deblurring model with bidirectional information complementarity. We introduce a content-aware motion magnitude attention module that utilizes learnable mask to extract relevant information from blurry images effectively, and we incorporate a transposed cross-attention fusion module to efficiently combine features from both spike data and blurry RGB images. Furthermore, we build two extensive synthesized datasets for training and validation purposes, encompassing high-temporal-resolution spikes, blurry images, and corresponding sharp images. The experimental results demonstrate that our method effectively recovers clear RGB images from highly blurry scenes and outperforms state-of-the-art deblurring algorithms in multiple settings.

## 1 Introduction

Traditional computer vision is well developed and models on visual tasks are satisfying [10, 37, 36, 11]. As a mature industrial product, frame-based cameras have also been fully developed whose imaging quality is high. Due to the existence of the exposure window, high-speed scenes or moving objects will cause serious blur effects as illustrated in Fig. 1(a), which degrades or invalidates the model performance. Thus, deblurring task remains an important demand for applications. With the development of convolutional neural networks and Transformers, models like Restormer [59] and SwinIR [32] used for deblurring tasks perform well. Nevertheless, our main concern is that the performance of these models is not reliable as the blurry image lacks sufficient contextual information to recover the corresponding clear image. When the blurry artifact is severe, models only input with blurry images may fail.

---

*Equal contributors.
†Corresponding authors.

37th Conference on Neural Information Processing Systems (NeurIPS 2023).

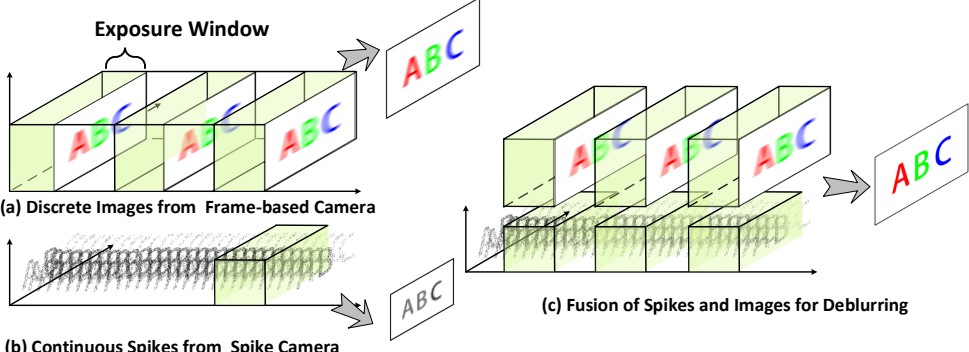

**(a) Discrete Images from Frame-based Camera**

**(b) Continuous Spikes from Spike Camera**

**(c) Fusion of Spikes and Images for Deblurring**

Figure 1: Illustration of the motivation of this paper. (a) The exposure window leads to motion blur in frame-based cameras. (b) Continuous spikes can recover low-resolution gray images yet with high temporal resolution. (c) Our method aims to fuse two modalities for motion deblurring.

Neuromorphic vision sensors, including event cameras [1, 33] and spike cameras [16], have gained recent attention due to advantages such as high temporal resolution, dynamic range, and low power consumption. Event cameras capture changes in light intensity, while spike cameras record absolute intensity in a unique way. They accumulate voltage converted from the light signal asynchronously and emit spikes when a preset threshold is reached, achieving a time resolution of up to 40,000Hz. Spike cameras are motion-sensitive and capture comprehensive texture information, offering the potential to mitigate motion blur. However, the spike stream is not directly interpretable by humans like RGB images. Initial research aimed at reconstructing grayscale images from spikes (see Fig. 1(b)). Subsequent studies leveraged spikes for high-level vision tasks, including object tracking, optical flow estimation, and depth estimation. Presently, spike cameras face challenges in these tasks due to their lower spatial resolution and absence of color. **We consider utilizing spike cameras to assist in image deblurring.** Our intuition is that the information in the RGB images and the neuromorphic data are complementary. The image domain contains rich spatial texture information with high-fidelity color, while the neuromorphic domain provides abundant temporal information [72, 9] promises them to record fast motions. Some studies utilize event cameras to assist image deblurring [57, 63, 49]. However, most event-based methods unidirectionally utilize information from the event domain to assist the image domain, without achieving the complementarity of information from both domains. Unlike event cameras, spike cameras maintain high temporal resolution while also recording texture information in the form of spike intervals [72]. The complete texture information, as opposed to solely motion edge information, can offer more valuable cues for deblurring tasks. As shown in Fig. 1(c), we aim to simultaneously consider the bidirectional transmission of information from both modalities, thereby achieving more efficient motion deblurring.

Therefore, we propose the first spike-based RGB image deblurring model named **SpkDeblurNet**. The model takes high-resolution blurry RGB images and low-resolution spike streams as input, each of which passes through one Transformer-based branch. We design a content-aware motion-magnitude attention module to fuse the high-resolution features in images into the spike branch. We further propose transposed cross-attention to perform a bidirectional fusion of cross-modal features between two branches. Moreover, we establish two high-quality datasets **Spk-X4K1000FPS** and **Spk-GoPro** for both training and validation. In experiments, we compare with various deblurring methods and prove the effective assistance of spikes. The complementarity of the two modalities is also verified. Our contributions to this work can be summarized as follows:

- We propose the first spike-based motion deblurring model with bidirectional information complementarity. Specifically, we propose a content-aware motion magnitude attention module based on learnable mask to utilize effective information in blurry images. A transposed cross-attention fusion module is built to efficiently fuse features from spikes and blurry RGB images.

- We generate two large-scale synthesized datasets used for training and validation, which contain high-temporal-resolution spikes, blurry images, and sharp images.

- Experimental results demonstrate that the proposed method achieves ideal results, outperforming other deblurring methods.

## 2 Related Works

**Research on Spike Camera.** Spike cameras possess the advantages of ultra-high temporal resolution and high dynamic range, which bring the potential to solve various visual tasks, especially in high-speed scenes. Early studies focus on recovering clear gray-scale images from spikes. Zhu *et al.* [72] propose the first work by directly analyzing the spike firing rate and the inter-spike interval. Some works [73, 69] take biological plausibility into consideration. Spk2imgnet [65] achieves satisfying performance by training deep convolutional neural networks (CNNs) in a supervised manner. Chen *et al.* [5] introduce the self-supervised framework. Besides, some studies [56, 64] explore recovering super-resolution images from spikes. Recently, more researchers have used spike cameras to solve various visual tasks. Zhao *et al.* [66] build a new spike-based dataset for object recognition. Zheng *et al.* [68] propose a bio-inspired framework for object tracking. Hu *et al.* [15] contribute to spike-based optical flow estimation by proposing the SCFlow net and new datasets. Zhao *et al.* [67] proposed to jointly estimate a series of flow fields. Chen *et al.*[6] propose a self-supervised framework for joint learning of optical flow and reconstruction. Besides, Spike Transformer [62] and SSDEFormer [53] are proposed for monocular/stereo depth estimation. Additionally, we observed similarities between single-photon cameras (SPCs) and spike cameras in their binary output. Nonetheless, SPCs predominantly depends on single-photon avalanche diode (SPAD) detector [35, 17, 27] or quanta image sensors (QIS) [38, 7, 2, 13], while the spike camera is based on CMOS sensors like traditional cameras and employs standard semiconductor manufacturing processes, rendering them fundamentally distinct, with the latter being more cost-efficient.

**Frame-Based Motion Deblurring.** Traditional frame-based motion deblurring methods typically employ optimization techniques [12, 24, 29, 30, 58] and blur kernel estimation [28, 55, 42, 21]. In recent years, with the advancement of deep learning, end-to-end approaches based on CNNs and transformers have demonstrated impressive performance in image deblurring. Multi-scale [39, 50], multi-stage [61, 4], and progressive strategies [8] have been proposed to improve the effectiveness of end-to-end training. Some works [25, 26] proposed to utilize Generative Adversarial Networks (GANs) to provide better human perception. On the other hand, universal networks [3, 31, 32, 54, 61, 59, 60, 4] have been introduced to address a series of low-level problems including image deblurring. Further research applied the powerful transformer framework [54, 31, 32, 59, 23] to motion debluring, achieving state-of-the-art (SOTA) performance. Unlike the conventional one-to-one mapping from a blurry image to a single sharp image, blurry frame interpolation or blur decomposition aims to recover multiple sharp images from a single blurry image [20, 19, 44, 45, 40, 70]. Additionally, directional ambiguity can be avoided in methods based on blurry video inputs [19, 44, 45, 40, 70] due to the motion cues of adjacent frames.

**Event-Based Motion Deblurring.** There has been increasing interest in utilizing event cameras to assist image deblurring. Several works [18, 43] utilized optical flow to provide motion information. NEST [51] proposed an efficient event representation based on bidirectional LSTM for deblurring. Kim *et al.* [22] extended the task to images with unknown exposure time. EFNet [48] introduced a multi-head attention module to better fuse event and image information. Some works [34, 41, 52, 49, 63, 57] have utilized events to assist blurry frame interpolation or blur decomposition. In addition, a series of high-quality event-image datasets have been proposed [51, 49, 18, 48], providing convenience for subsequent research. Most event-based methods merely utilize event-based information to assist the image domain task. In this paper, we aim to use the bidirectional transmission of information from both modalities to achieve more efficient motion deblurring.

## 3 Methods

### 3.1 Spike Generation Mechanism

Spike cameras, different from the differential sampling of event cameras, adopt the integral sampling mechanism. A spike camera with the spatial resolution of $H \times W$ is composed of an array of units on each pixel. The working mechanism of the spike camera is illustrated in Fig. 2. For a unit at pixel $(x, y)$ that $x \in [0, W), y \in [0, H)$ continuously receives coming photons at any timestamps $t$ and converts the light intensity $L_{x,y}(t)$ into the electric current, and increases its voltage $V_{x,y}(t)$. A voltage threshold $\Theta$ is pre-defined. A spike will be triggered once the threshold is reached, and the $V_{x,y}(t)$ will be reset to 0. The process can be formulated as follows:

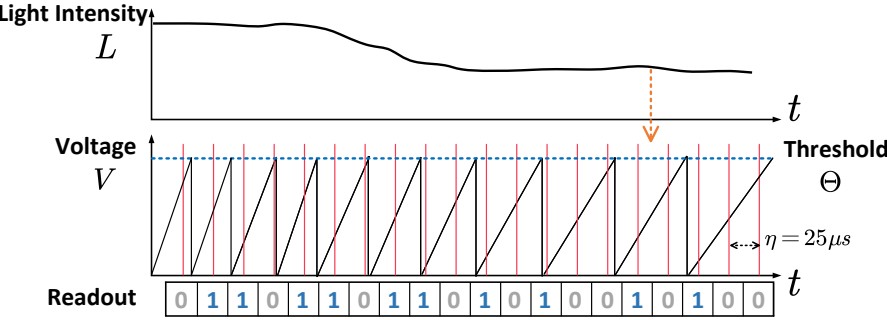

Figure 2: Illustration of the mechanism of generating spikes as the light intensity changes.

$$V_{x,y}^+(t) = \begin{cases} V_{x,y}^-(t) + \alpha \cdot L_{x,y}(t), & \text{if } V_{x,y}^-(t) < \Theta, \\ 0, & \text{otherwise,} \end{cases} \tag{1}$$

where $V_{x,y}^-(t)$ and $V_{x,y}^+(t)$ denotes the voltage before and after receiving the electric current, and the $\alpha$ is the photoelectric conversion factor. The back-end circuit reads out spikes continuously with a very small interval $\eta$ ($25\mu$s). A pixel at $(x, y)$ will output $S(x, y, n) = 1$ when the threshold is reached at $t \in (\eta(n - 1), \eta n]$, otherwise output $S(x, y, n) = 0$. Thus, for a $T$-times readout, a spike train with the size of $H \times W \times T$ would be generated.

### 3.2 Problem Statement

A spike camera outputs binary spike streams with very-high temporal resolution. A conventional frame-based camera outputs an image sequence and each image is generated after an exposure time $\mathbf{T}_e$. Though scenes in the world are continuous, images are discrete. We regard spikes approximately as continuous signals. When coming to scenarios with high-speed motions, the exposure window in frame-based cameras leads to a blurred image $\mathbf{B}_e$, while spikes $\mathbf{S}_e$ during the exposure time finely records the motion dynamics. In this work, we aim to utilize the spike stream $\mathbf{S}_e$ during the $\mathbf{T}_e$ to deblur the RGB image $\mathbf{B}_e$. Moreover, the current spatial resolution of spike cameras is relatively low, so we set the spatial resolution of $\mathbf{S}_e$ to half of $\mathbf{B}_e$ to simulate the real situation.

### 3.3 Architecture

To achieve effective integration of the high resolution color texture information in conventional images and the high temporal resolution offered by spike streams, we present a novel two-branch fusion spike-based image deblurring framework, *SpkDeblurNet* (Fig. 3). Our framework combines the benefits of both image deblurring and spike-based image processing. In the first branch of SpkDeblurNet, we input the blurred RGB image $\mathbf{B}_e$, while the second branch takes a continuous spike stream $\mathbf{S}_e$ centered around the latent clear image. $\mathbf{B}_e$ has twice the spatial resolution of $\mathbf{S}_e$.

The input blurry image $\mathbf{B}_e^{H \times W \times 3}$ is first passed through two $3 \times 3$ convolutional layers with a stride of 2 for downsampling. Subsequently, the downscaled features are fed into a series of continuous Residual Swin Transformer blocks (RSTB) [32, 37]. These RSTBs play a crucial role in extracting features that encapsulate enhanced spatiotemporal semantics. Mathematically, we can express this process as follows:

$$f_1^b = \mathbf{RSTB}_1^b(\mathbf{F}_{down}^b(\mathbf{B}_e^{H \times W \times 3})), \tag{2}$$

$$f_{i=2,3}^b = \mathbf{RSTB}_i^b(f_{i-1}^b), \tag{3}$$

where the $\mathbf{F}_{down}^b(\cdot)$ denotes the downsampling operations.

Similarly, the spike stream $\mathbf{S}_e^{H \times W \times T}$ undergoes processing through a singular convolutional layer, facilitating the embedding of spike streams within the feature domain. Furthermore, a sequence of

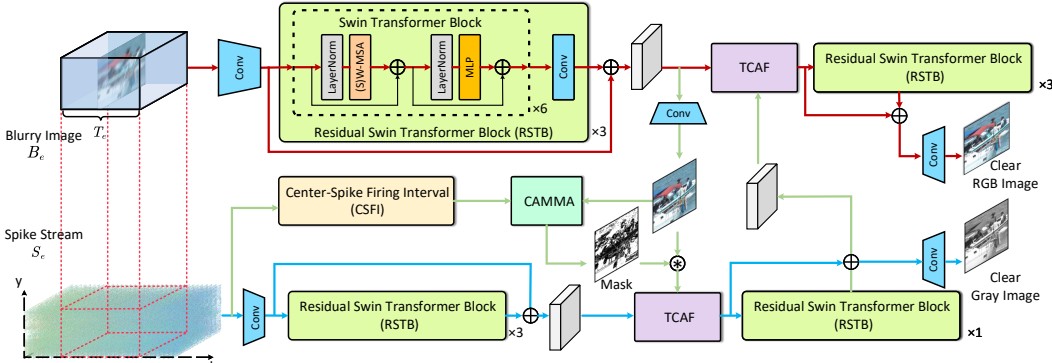

Figure 3: The architecture of the SpkDeblurNet. Arrows in red denote the image branch, the blue denotes the spike branch, and the green denotes the bidirectional cross-model feature fusion process.

three RSTBs is employed to execute profound spatiotemporal semantic extraction:

$$f_1^s = \mathbf{RSTB}_1^s(\mathbf{F}_{down}^s(\mathbf{S}_e^{H \times W \times T})), \tag{4}$$

$$f_{i=2,3}^s = \mathbf{RSTB}_i^s(f_{i-1}^s). \tag{5}$$

To effectively merge the comprehensive high resolution color texture information available in the image domain with the spatiotemporal semantic extraction performed on the spike stream, we design a novel Content-Aware Motion Magnitude Attention module (CAMMA). This module facilitates the extraction of features specifically from the regions of clarity within the initial deblurred image, thereby producing a robust motion mask represented as $\mathcal{M}$. By combining this motion mask with the initial deblurred image $\mathbf{F}_{up}^{'b}(f_3^b)$ extracted from the blurry image and integrating it with the feature $f_3^s$ obtained from the spike domain using the Transposed Cross-Attention Fusion (TCAF) module, a clear grayscale image denoted as $\hat{I}_t^g$ is generated through an RSTB and a post-upsampling process. The mathematical expression for this procedure is as follows:

$$f_{tcaf}^s = \mathbf{TCAF}^s(\mathcal{M} * \mathbf{F}_{up}^{'b}(f_3^b), f_3^s), \tag{6}$$

$$\hat{I}_t^g = \mathbf{F}_{up}^s(\mathbf{RSTB}_4^s(f_{tcaf}^s)), \tag{7}$$

with $*$ denotes the element-wise multiplication, and $\hat{I}_t^g$ denotes the predicted grayscale image at the center timestamp $t$ of window $\mathbf{T}_e$.

We then combine the spike features $f_4^s$ with the feature information $f_3^b$ obtained by the blurry image through the TCAF module, and output a refined sharp color image $\hat{I}_t$ with a series of RSTBs and upsampling, which can be summarized as follows:

$$f_{tcaf}^b = \mathbf{TCAF}^b(f_4^s, f_3^b), \tag{8}$$

$$\hat{I}_t = \mathbf{F}_{up}^b(\mathbf{RSTBs}^b(f_{tcaf}^b)), \tag{9}$$

In the blurry decomposition task, a common issue is directional ambiguity [70, 71]. Since we reconstruct the center position of the exposure window in the spike branch, the output of $\mathbf{RSTB}^s$ contains time-specific spike encoding information, which can avoid this problem. During online inference, we can decompose the blurry image into different timestamps by moving the position of the spike stream, and the frame rate of the decomposed images depends on the sampling rate of the spike camera.

### 3.4 Cross-Modal Information Complementing

**Spike Reconstruction Based on High Resolution Prior in Image Domain.** Each pixel in the spike camera asynchronously outputs high-time-resolution spikes according to the brightness proportional to the input light intensity. This mechanism enables the spike stream to effectively capture and represent both temporal and spatial information of visual inputs. However, the current spatial resolution of the

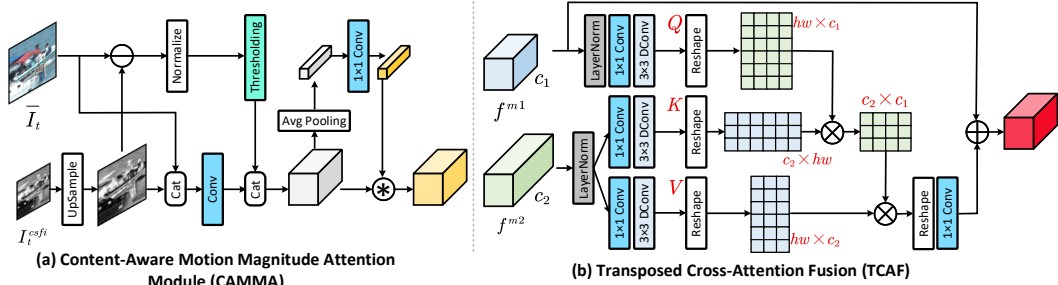

**(a) Content-Aware Motion Magnitude Attention Module (CAMMA)**

**(b) Transposed Cross-Attention Fusion (TCAF)**

Figure 4: Illustration of the Content-Aware Motion Magnitude Attention Module and the Transposed Cross-Attention Fusion Module.

spike camera is relatively low. In our module, we explicitly reconstruct high-resolution grayscale images of the spike stream to enable the network to learn image-like features. Given the ground truth grayscale image $I_t^g$ at the moment of the exposure window center, the loss function is formulated as:

$$\mathcal{L}_{recon} = \|I_t^g - \hat{I}_t^g\|_1, \tag{10}$$

Reconstructing high-resolution frames solely from the spike stream itself remains challenging. To fully leverage the advantages of both input modalities, we propose to utilize information from the image domain as a high-resolution prior to spike reconstruction. However, the input images are mostly highly blurred, which hinders the provision of high-resolution clues for reconstruction. Therefore, we further introduce the CAMMA module to adaptively search for regions with smaller motion magnitudes in the initial deblurred image, and fuse them into the spike branch.

**Content Aware Motion Magnitude Attention Module.** In order to incorporate high-resolution prior knowledge from the image domain into the spike domain, an intuitive idea is to directly fuse the input blurred image with the spike branch. The blurry foreground caused by moving objects cannot provide effective texture information, which can be handled by using a mask to retain sharp backgrounds [47]. Moreover, the global blur caused by camera motion is unable to provide effective high-resolution prior. Therefore, we propose a CAMMA module to encode the motion magnitude from the initial deblurred result. As shown in Fig. 4, we first pass the $f_3^b$ through shallow convolution and upsampling to obtain the initial deblurred image $\overline{I}_t$. Through training, the blur in $\overline{I}_t$ is initially reduced. Subsequently, we need a coarse sharp image as a reference to obtain the blurry region by comparing it with $\overline{I}_t$. We adopt the basic statistical method text-from-interval (TFI) [72] to center-spike firing interval (CSFI) to obtain the central coarse sharp image. CSFI estimates the pixel value of the current pixel based on the observation that the spike camera's firing frequency at each pixel is proportional to the real light intensity by statistically analyzing the spike interval. The process is formulated as:

$$I_t^{csfi} = \frac{C}{\Delta t}, \tag{11}$$

where $\Delta t$ represents the inter-spike interval corresponding to moment $t$, and the $C$ refers to the maximum dynamic range of the reconstruction. Subsequently, we perform differencing and normalization on the upsampled $I_t^{csfi}$ and $\overline{I}_t$ and then apply thresholding to obtain the hard thresholding motion magnitude mask. However, since $I_t^{csfi}$ is merely a basic reconstruction and suffers from noise [72]. In order to enable the network to learn a more robust motion magnitude mask, we concatenate $I_t^{csfi}$ and $\overline{I}_t$ and use simplified channel attention [3] with the hard thresholding mask to obtain a content-aware motion magnitude mask $\mathcal{M}$. After multiplying the motion magnitude mask with $\overline{I}_t$, we utilize the TCAF module to fuse it into the spike branch. We refer to the CAMMA module, the CSFI process, and the $\mathbf{F}_{up}^{'b}(\cdot)$ module as the CAMMA branch, with which the high-resolution prior knowledge from the image domain can be transferred effectively into the spike domain.

**Spike-Guided Motion Deblurring.** As mentioned in Sec. 3.3, our SpkDeblurNet involves information transfer from image to spike and spike to the image. The modalities differ significantly for the former, while the differences are minor for the latter. In order to better integrate the two modalities, we propose the TCAF Module. The query ($\mathbf{Q}^{m1}$) is derived from one modality $m1$ with $c_1$ channels, while the key ($\mathbf{K}^{m2}$) and value ($\mathbf{V}^{m2}$) are obtained from another modality $m2$ with $c_2$ channels. The attention is computed using the following equation:

$$\hat{f}^{m1} = f^{m1} + \mathbf{V}^{m2} \cdot \mathbf{G}(\mathbf{K}^{m2} \cdot \mathbf{Q}^{m1}/\alpha), \tag{12}$$

Table 1: Comparison of various motion deblurring methods on Spk-X4K1000FPS.

| Method | Extra Data | PSNR ↑ | SSIM ↑ | PSNR ↑ | SSIM ↑ | #Param |
|---|---|---|---|---|---|---|
| **Spk-X4K1000FPS** | | $e = 33$ | | $e = 65$ | | |
| HINet [4] | - | 33.98 | 0.933 | 29.19 | 0.867 | 88.7M |
| NAFNet [3] | - | 34.13 | 0.937 | 29.06 | 0.878 | 67.9M |
| EFNet [48] | Spike | 36.36 | 0.960 | 33.53 | 0.937 | 8.5M |
| REFID [49] | Spike | 36.30 | 0.962 | 33.47 | 0.945 | 15.9M |
| **SpkDeblurNet (Ours)** | Spike | **37.42** | **0.968** | **35.94** | **0.966** | 13.5M |

Where $f^{m1}$ and $\hat{f}^{m1}$ are the features before and after fusion for modality $m1$, $\mathbf{G}$ denotes the softmax function and $\alpha$ is a learnable scaling parameter. However, the calculation of attention faces two problems: 1) The spike branch is lightweight with fewer channels compared to the deblurring branch, which makes traditional attention computation in the spatial dimension not directly applicable. 2) The computational cost of traditional attention grows quadratically with the input size, which is not acceptable for image restoration tasks. In the main network, we mitigate these issues by employing the RSTB [37, 36], which calculates attention within windows to reduce computation. In TCAF, we adopt the Transposed Attention proposed in [59], which performs attention computation along the channel dimension, thereby addressing the aforementioned problems. We apply the TCAF module to the fusion of features in both directions.

## 3.5 Joint Learning of Deblurring and Reconstruction

We jointly train the proposed SpkDeblurNet. The spike reconstruction branch obtains high-resolution image priors from the initial deblurred result, enabling the recovery of better image-like features. The features from the spike branch in turn guide the deblurring branch, providing time-specific texture information. For both the initial deblurred result $\overline{I}_t$ and the refined deblurred result $\hat{I}_t$, we utilize the L1 loss for supervision during training. The loss function of the entire network can be formulated as follows:

$$\mathcal{L} = \|I_t - \hat{I}_t\|_1 + \lambda_1 \|I_t^g - \hat{I}_t^g\|_1 + \lambda_2 \|I_t - \overline{I}_t\|_1, \tag{13}$$

where $\lambda_1$ and $\lambda_2$ are hyperparameters that control the loss terms.

## 3.6 Datasets

This paper is the first study on spike-based motion deblurring, and no large-scale training datasets are currently available. Therefore, we built two large-scale spike-based datasets based on existing real image datasets [39, 46] for training and validation. The first one, named 'Spk-X4K1000FPS', is built on the dataset X4K1000FPS which contains clear images captured by 1000fps cameras. To generate spikes, we utilize the interpolation method [46] to further increase the frame rate four times. Then we build a simple spike simulator according to the spike generation mechanism. It inputs continuous sharp images as light signals and converts them to spikes. For clips in the dataset, we use the average of neighboring $e$ sharp images around the ground truth images to generate blurry images, where $e = 33, 65$. The second one, named 'Spk-GoPro' is built on the GoPro [39] dataset in a similar way as above. More implementation details are attached in the supplementary materials.

# 4 Experiments

## 4.1 Comparative Results

In this section, we present comparative results on both Spk-X4K1000FPS and Spk-GoPro datasets. For Spk-X4K1000FPS, we generate blurry images by setting an exposure window of $e = 33$ as well as an extreme exposure window of $e = 65$. For Spk-GoPro, we use the provided blurry images. Implementation details can be found in supplementary materials.

**Results on Spk-X4K1000FPS.** Tab. 1 and Fig. 5 present quantitative and qualitative comparisons of our method with other approaches under two different exposure window settings. For EFNet and REFID, we use their networks for spike-assisted deblurring. For EFNet, we employed input representations similar to its SCER approach, excluding the EMGC module due to its event-related nature. Regarding REFID, we utilized similar spike voxel representations. All comparative methods

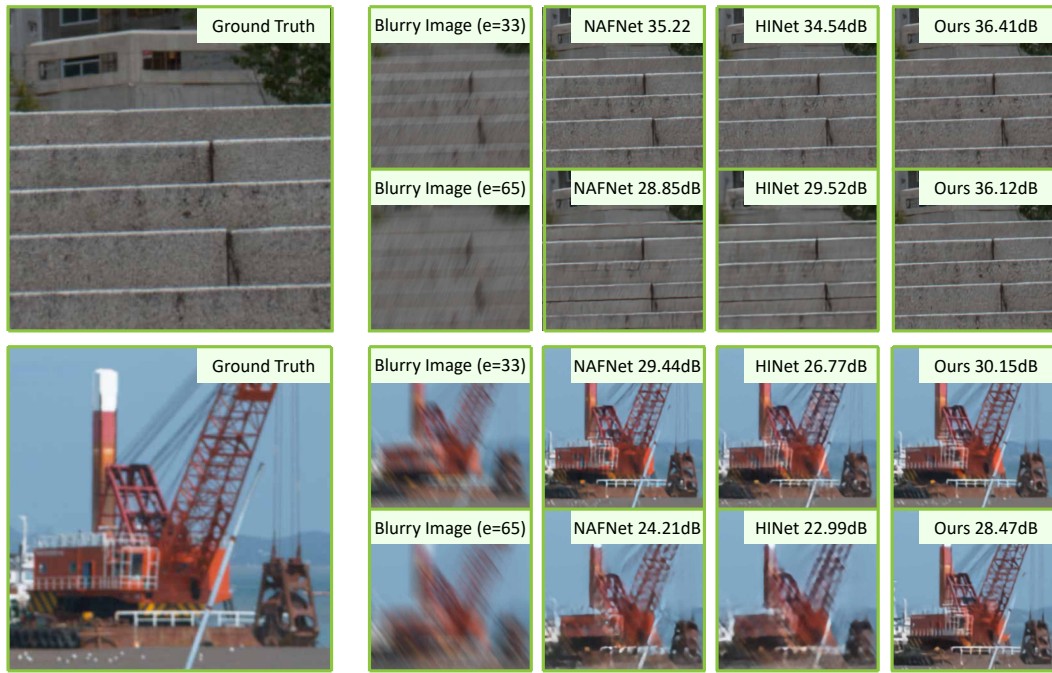

Figure 5: Visualized results on Spk-X4K1000FPS dataset, compared with NAFNet [3] and HINet [4].

were trained from scratch on the proposed dataset. We observe that both image-domain methods perform well in recovering image details with a smaller exposure window setting ($e = 33$). However, when the exposure window becomes larger ($e = 65$), pure image-based methods struggle to find cues from extremely blurry scenes to restore clear images, resulting in significant loss of texture details or generating artifacts. In the second row of the first sample sequence in Fig. 5, NAFNet and HINet mistakenly recover non-existent stair edges. In contrast, our multimodal approach leverages spike streams to find cues and restore correct and clear textures. Additionally, the networks in event-assisted methods fall short of our network. This is due to our network's superior bidirectional information transmission nature.

**Results on Spk-GoPro.** As shown in Tab. 2, our proposed SpkDeblurNet outperforms the state-of-the-art model REFID [49] and other competitive models in terms of PSNR On the GoPro dataset. We have observed that both our SpkDeblurNet and the SOTA event-based methods have significantly outperformed image-based methods, which demonstrates the substantial potential of neuromorphic data in complementing information for the image domain. Fig. 5 visualizes the deblurred results of different methods, from which we demonstrate that the quality of sharp images predicted by the SpkDeblurNet is more promising and satisfying than others. In the first sample of Fig. 5, our approach successfully restored the clear license plate number while also effectively recovering the fallen leaves on the ground, while other methods only achieved a relatively blurred restoration of the ground. Furthermore, our method excelled in restoring the clearest facial features of the person in the second sample. Zoom in for more details. Our framework, by leveraging the complementary information from both image and spike domains, has surpassed event-based methods that solely rely on unidirectional information transfer in terms of both quantitative and qualitative results.

**Results on Real-World Scenes.** To validate the generalization of our method in real-world scenarios, we construct a hybrid camera system to capture real-world spike-image data. We use a beam splitter to connect a spike camera and a conventional frame-based camera, enabling both cameras to have the same view. We rapidly wave objects in front of the cameras to simulate high-speed motion. We synchronize the two cameras and manually aligned their spatial and temporal outputs. Fig. 7 presents visual results of applying our model trained on the Spk-X4K1000FPS dataset with a window size of 33 to two real-world sequences. We can observe desirable results with clear textures and accurate colors. This demonstrates the good generalization of our proposed algorithm and dataset, highlighting the potential of our algorithm for real-world applications. More results and the hybrid camera system diagram can be found in the supplementary materials.

Table 2: Comparison of various motion deblurring methods on Spk-GoPro. HINet+: event-enhanced versions of HINet [4].

| Method | Extra Data | PSNR ↑ | SSIM ↑ | #Param |
|---|---|---|---|---|
| **Spk-GoPro** | | | | |
| DeblurGAN-v2 [26] | | 29.55 | 0.934 | - |
| D$^2$Nets [43] | Event | 31.60 | 0.940 | - |
| LEMD [18] | Event | 31.79 | 0.949 | - |
| MPRNet [61] | | 32.66 | 0.959 | 20.0M |
| HINet [4] | | 32.71 | 0.959 | 88.7M |
| Restormer [59] | | 32.92 | 0.961 | 26.1M |
| ERDNet [14] | Event | 32.99 | 0.935 | - |
| HINet+ [4] | Event | 33.69 | 0.961 | 88.7M |
| NAFNet [3] | | 33.69 | 0.967 | 67.9M |
| EFNet [48] | Event | 35.46 | 0.972 | 8.5M |
| REFID [49] | Event | 35.91 | **0.973** | 15.9M |
| **SpkDeblurNet (Ours)** | Spike | **36.12** | 0.971 | 13.5M |

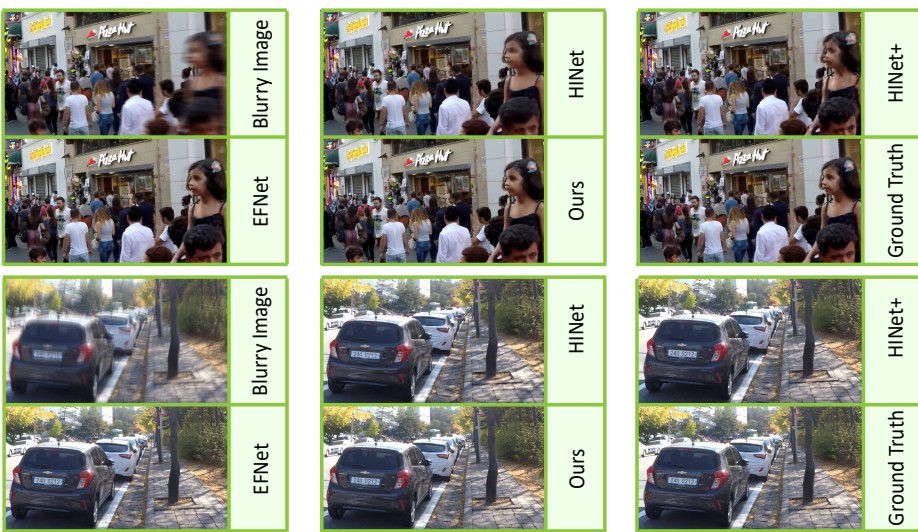

Figure 6: Visualized results on Spk-GoPro dataset.

## 4.2 Ablation Studies

In this section, we conduct a series of ablation studies on Spk-X4K1000FPS dataset.

**Effectiveness of the proposed modules.** As shown in Tab. 3, removing the CAMMA branch resulted in a performance degradation of 0.2-0.3 dB, confirming the necessity of introducing features from the deblurring branch and the effectiveness of the CAMMA module's learnable magnitude masking mechanism. Additionally, replacing the TCAF module with concatenation led to minor performance degradation, which we attribute to the explicit supervision of the spike reconstruction branch enabling the branch to learn image-like features, thereby the modality gap is reduced.

**Information supplementation from image domain to spike domain.** We further conduct ablation studies on spike reconstruction with different combinations of three input representations and the usage of the CAMMA branch. To be specific, we employ either coarse reconstruction CSFI or a symmetric cumulative spike representation similar to EFNet [48]. For the latter, we accumulate spikes from the center towards both ends using three window lengths and concatenate the results along the channel dimension. As shown in Tab. 4, CAMMA consistently significantly improves spike reconstruction across all input representations. For settings using CSFI and cumulated spikes as inputs, CAMMA enhances performance by 1.5-3.5 dB. This is due to the simpler nature of CSFI and cumulated spikes, which contain less texture and motion information. High-resolution image priors from the deblurring branch compensate for this, further enhancing spike reconstruction. Even with spike streams as input, CAMMA still yields improvements of 0.4-0.7 dB. This set of ablation experiments demonstrates the effectiveness of the proposed scheme in incorporating high-resolution features from the deblurring branch into the spike branch, thereby enhancing spike reconstruction performance and providing better guidance and cues for the deblurring task.

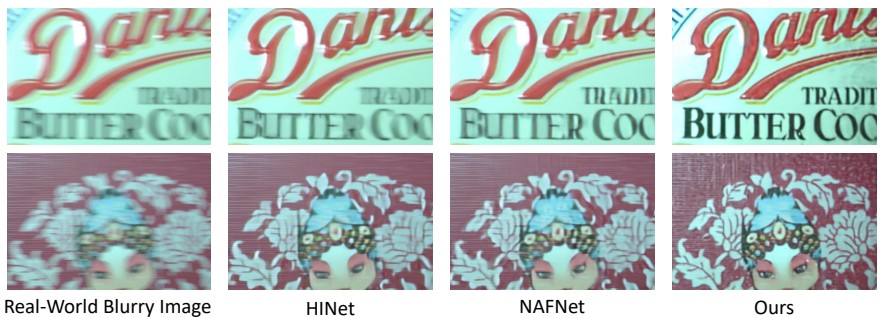

| Real-World Blurry Image | HINet | NAFNet | Ours |

Figure 7: Qualitative Results on Real-World Scenes.

Table 3: Ablation study of our proposed modules and input representations.

| Experiment settings | PSNR ↑ | SSIM ↑ | PSNR ↑ | SSIM ↑ |
|---|---|---|---|---|
| **Spk-X4K1000FPS** | $e = 33$ | | $e = 65$ | |
| Replace input representation with CSFI | 36.08 | 0.957 | 34.22 | 0.950 |
| Replace input representation with cumulate spikes | 36.34 | 0.955 | 34.92 | 0.955 |
| Replace TCAF with concatenation | 37.36 | 0.968 | 35.81 | 0.965 |
| Remove CAMMA branch | 37.20 | 0.967 | 35.64 | 0.965 |
| **Our final SpkDeblurNet** | **37.42** | **0.968** | **35.94** | **0.966** |

Table 4: Ablation study of CAMMA on **spike reconstruction** with different inputs.

| Experiment settings | | PSNR ↑ | SSIM ↑ | PSNR ↑ | SSIM ↑ |
|---|---|---|---|---|---|
| Input Representation | CAMMA | $e = 33$ | | $e = 65$ | |
| CSFI | ✗ | 33.67 | 0.923 | 33.69 | 0.922 |
| CSFI | ✔ | **37.25** | **0.939** | **36.13** | **0.935** |
| Cumulated Spikes | ✗ | 35.58 | 0.926 | 35.11 | 0.914 |
| Cumulated Spikes | ✔ | **37.54** | **0.935** | **36.65** | **0.936** |
| Spike Streams | ✗ | 38.72 | 0.953 | 38.91 | 0.952 |
| **Spike Streams** | ✔ | **39.41** | **0.955** | **39.32** | **0.954** |

**Information supplementation from spike domain to image domain.** We investigate the complementary role of the spike domain to the image domain by replacing the input representation. Quantitative results in Tab. 3 indicated that using coarse reconstruction as input still yielded competitive performance, suggesting that the rough texture information contained in CSFI serves as a good complement to the deblurring branch. On the other hand, using cumulated spikes as input further improved the performance. As a comparison, we achieved the highest performance using pure spike stream as input, demonstrating the network's ability to fully excavate the rich temporal information and clues embedded in spike streams to recover clear images. We further conduct ablation studies on different combinations of three input representations and the usage of the CAMMA module for the deblurring task, the results can be found in the supplementary materials.

## 5 Conclusion

In this paper, we propose the first spike-based motion deblurring model with bidirectional information complementarity, in which we propose a content-aware motion magnitude attention module and a transposed cross-attention fusion module. We build two large-scale synthesized datasets used for both training and validation, which contain high-temporal-resolution spikes, blurry images, and sharp images. Experimental results demonstrate that the proposed method outperforms other deblurring methods and achieves desirable results.

**Limitations.** Our future work will prioritize enhancing the resolution of the spike stream to improve its auxiliary capability for image-domain tasks, considering the limitation of the current low resolution of spike cameras and the significant impact of spatial super-resolution on applications.

## Acknowledgments

This work was supported by the National Natural Science Foundation of China under Grant No. 62176003 and No. 62088102.

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
