# Supplementary Material for *Enhancing Motion Deblurring in High-Speed Scenes with Spike Streams*

**Shiyan Chen**[1,2*] **Jiyuan Zhang**[1,2*] **Yajing Zheng**[1,2†] **Tiejun Huang**[1,2,3] **Zhaofei Yu**[1,2,3†]

[1]School of Computer Science, Peking University
[2]National Key Laboratory for Multimedia Information Processing, Peking University
[3]Institute for Artificial Intelligence, Peking University
{strerichia002p,jyzhang}@stu.pku.edu.cn,
{yj.zheng,yuzf12,tjhuang}@pku.edu.cn

## A    Implementation Details

We train SpkDeblurNet on settings with blur windows $e = 33$ and $e = 65$. In the deblurring branch, the RSTB blocks input with features which have $\frac{1}{4}$ size from the input blurry image $\mathbf{B}_e^{H \times W \times 3}$. In the spike reconstruction branch, the spike stream $\mathbf{S}_e^{\frac{H}{2} \times \frac{W}{2} \times T}$ is first down-sampled by a factor of two through a convolutional layer, so that the spikes and images are of the same size in the feature domain. All RSTB blocks consist of 6 STB blocks. The values of $\lambda_1$ and $\lambda_2$ are set to 1 based on empirical observations. To prepare the input image and spikes, we utilize the same seed for random cropping, generating patches of size $256 \times 256$ and $128 \times 128$ respectively. Each GPU processes a batch size of 8. Data augmentation techniques such as random flipping and rotation are employed. We use AdamW [3] as the optimizer, setting the learning rate and weight decay to 1e-4. We train for a total of 1e5 iterations, use a cosine scheduler to schedule the learning rate, and ensure that the learning rate minimum is not lower than 1e-6. The quantitative metrics employed for evaluation are peak signal-to-noise ratio (PSNR) and structural similarity (SSIM).

## B    Datasets Details

For the Spk-X4K1000FPS dataset, we follow the data partitioning scheme of X4K1000FPS [5]. The training set consists of 4,408 clips of size $768 \times 768$, with most clips containing 65 consecutive frames. The test set comprises 15 sequences of size $512 \times 512$, each with varying degrees of occlusion, optical flow magnitudes, and scene diversity. Each sequence contains 33 frames. We use an interpolation algorithm [5] to further increase the frame rate fourfold, and select center frames with equal interval as ground truth. Blurry images with different motion magnitudes are generated by averaging the surrounding 33 or 65 images. For spike data, we first downsample the interpolated dataset and then used a spike camera simulator to generate the corresponding spike streams. The photoelectric transformation coefficient of the simulator is set to 0.5.

For the Spk-GoPro dataset, to address the low frame rate of its original data [4], we applied a frame interpolation algorithm [5] to increase the frame rate by 8 times. To ensure a fair comparison with other methods, we utilized GoPro's native blurred images and ground truth data, as well as maintained the same data division. Specifically, the training set comprises 2,103 pairs of blurred images and their corresponding sharp images, both sized at $1,280 \times 720$. The test set consists of $1,111$ such image pairs. Furthermore, since the GoPro dataset generates blurred images by averaging 7-13 consecutive frames, we selected the smallest window size to generate the spike stream. Consequently, the exposure window for the generated spikes is 56.

---

[*]Equal contributors.
[†]Corresponding authors.

37th Conference on Neural Information Processing Systems (NeurIPS 2023).

Table S1: **Ablation Study of CAMMA on Deblurring with Different Input Representations.**

| Experiment settings | | PSNR ↑ | SSIM ↑ | PSNR ↑ | SSIM ↑ |
|---|---|---|---|---|---|
| Input Representation | CAMMA | $e = 33$ | | $e = 65$ | |
| CSFI | ✗ | 35.96 | 0.956 | 34.00 | 0.950 |
| CSFI | ✔ | **36.08** | **0.957** | **34.22** | **0.950** |
| Cumulated Spikes | ✗ | 36.00 | 0.951 | 33.52 | 0.935 |
| Cumulated Spikes | ✔ | **36.34** | **0.955** | **34.92** | **0.955** |
| Spike Streams | ✗ | 37.20 | 0.967 | 35.64 | 0.965 |
| **Spike Streams** | ✔ | **37.42** | **0.968** | **35.94** | **0.966** |

Table S2: **Comparisons between different methods regarding computational complexity.**

| Method | MACs | Params | Inference Time | PSNR (on GoPro) |
|---|---|---|---|---|
| HINet [2] | 170.49 G | 88.67 M | 20.2 ms | 32.71 dB |
| NAFNet [1] | 63.06 G | 67.79 M | 27.8 ms | 33.69 dB |
| EFNet [6] | 107.93 G | 7.73 M | 14.9 ms | 35.46 dB |
| REFID [7] | 4.36 T | 88.81 M | 781.2 ms | 35.91 dB |
| **Ours (e=65)** | 53.25 G | 12.93 M | 140.6 ms | N/A |
| **Ours (e=33)** | 53.18 G | 12.92 M | 107.9 ms | N/A |
| **Ours (e=56)** | 53.23 G | 12.93 M | 130.1 ms | 36.12 dB |

## C    Additional Ablation Studies of CAMMA

We further conduct ablation studies on different combinations of three input representations and the usage of the CAMMA branch for the deblurring task on Spk-X4K1000FPS dataset. As shown in Tab. S1, we observe that the introduction of CAMMA also improves the performance of deblurring across all settings. The performance gains range from 0.1 to 1.4 dB. This indicates that our CAMMA branch enhances the performance of the deblurring branch by improving the spike branch, thus demonstrating the effectiveness of the bidirectional complementary approach of the two modalities proposed in this paper.

## D    Complexity analysis

We have added comparisons regarding computational complexity and inference time in Tab. S2 (note that the settings for e=33 and e=65 are only applicable to the Spk-X4K1000FPS dataset). Our approach achieves a favorable balance between complexity and performance, as evident from our results.

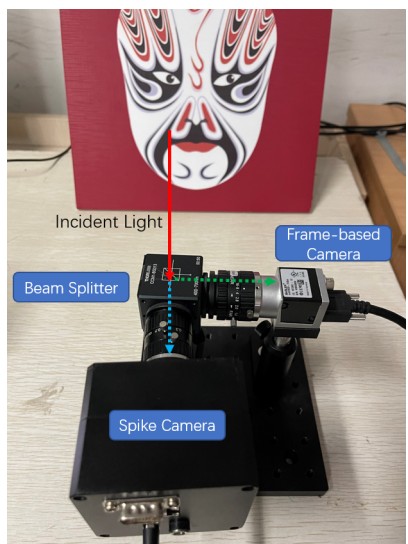

Figure S1: The prototype of our hybrid camera system.

## E    Additional Qualitative Results

### E.1    Additional Qualitative Results on Real-World Scenes

Fig. S1 presents our hybrid camera system. More qualitative results on real-world scenes can be found in Fig. S2. We can find that our method in the third sequence will make the scene darker, and we attribute it to the spike camera's sensitivity to ambient light. We will explore this issue in future research.

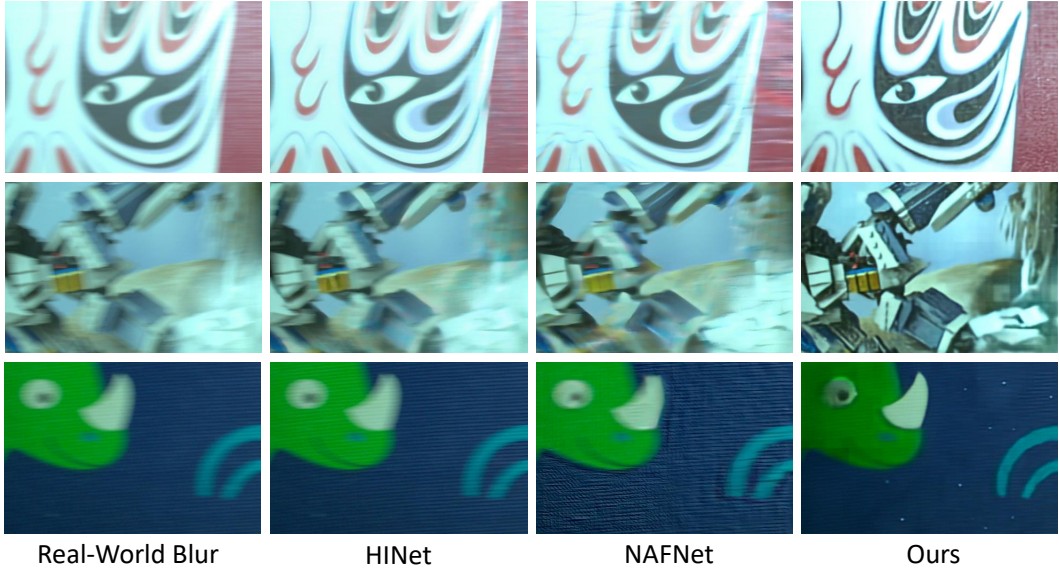

| Real-World Blur | HINet | NAFNet | Ours |

Figure S2: More Qualitative Results on Real-World Scenes.

## E.2 Extend to Blur Decomposition

As mentioned in Sec.3.3 of the main paper, we can extend our algorithm to the blur decomposition task. Since our algorithm recovers the central scene of the spike stream from the blurry image, our network can decompose a blurry image into different timestamps by shifting the spike stream in the temporal axis, and the frame rate of the blur decomposition depends on the frame rate of the spike stream. The temporal cues provided by the spike stream also mitigate the directional ambiguity issues [8, 9] commonly encountered in blur decomposition tasks. Fig. S3 illustrates the blur decomposition results of two sequences under exposure window $e = 65$, where we can observe that the blurry image is successfully decomposed into several consecutive and sharp images. Furthermore, even with slight misalignment in spatial and temporal domains after shifting the spike stream in the temporal axis, our algorithm is still able to accurately recover the sharp image at the specified timestamp, which demonstrates the robustness of our algorithm in handling the alignment problem between the two modalities.

## E.3 More Qualitative Results on Synthetic Datasets

In this section, we present additional qualitative results on synthetic datasets. Fig. S4 shows the qualitative results on Spk-X4K1000FPS, where we also showcase the spike reconstruction results. We can observe that although state-of-the-art methods such as NAFNet [1] and HINet [2] perform well at $e = 33$, their performance significantly deteriorates when $e = 65$. On the other hand, our method is able to restore clear textures under both exposure windows. Fig. S5 illustrates the qualitative results on Spk-GoPro dataset, our method preserves more details. Zoom in to examine more details.

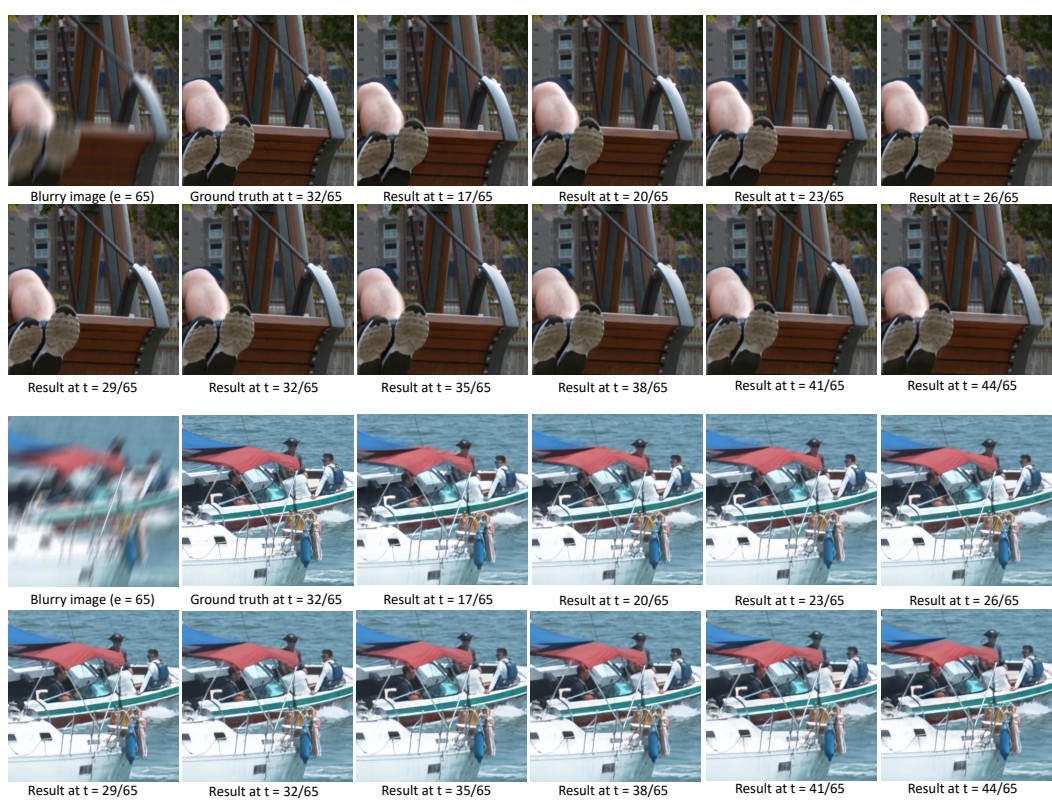

Figure S3: Blur Decomposition Results on Spk-X4K1000FPS Dataset.

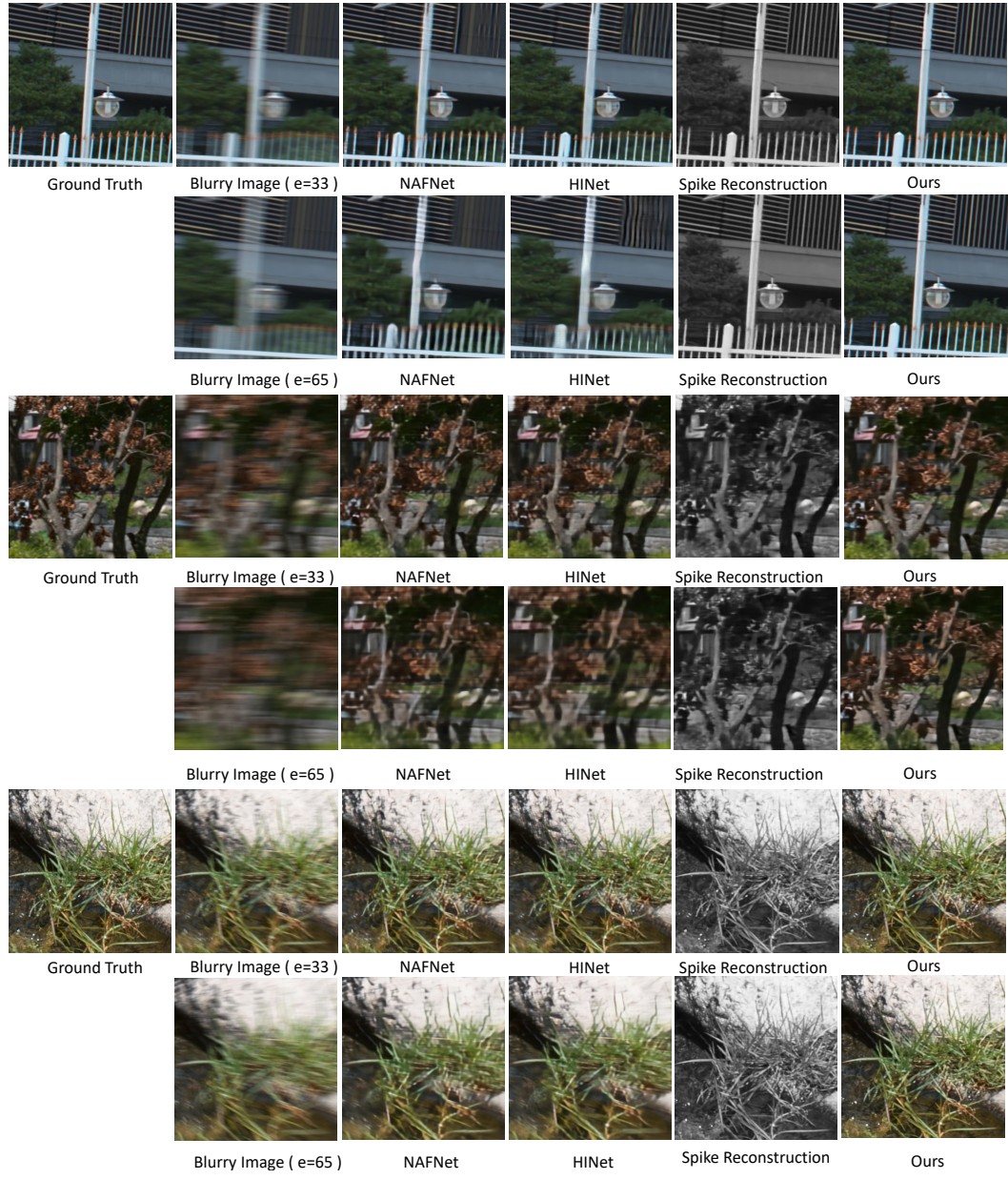

Figure S4: More Qualitative Results on Spk-X4K1000FPS Datasets.

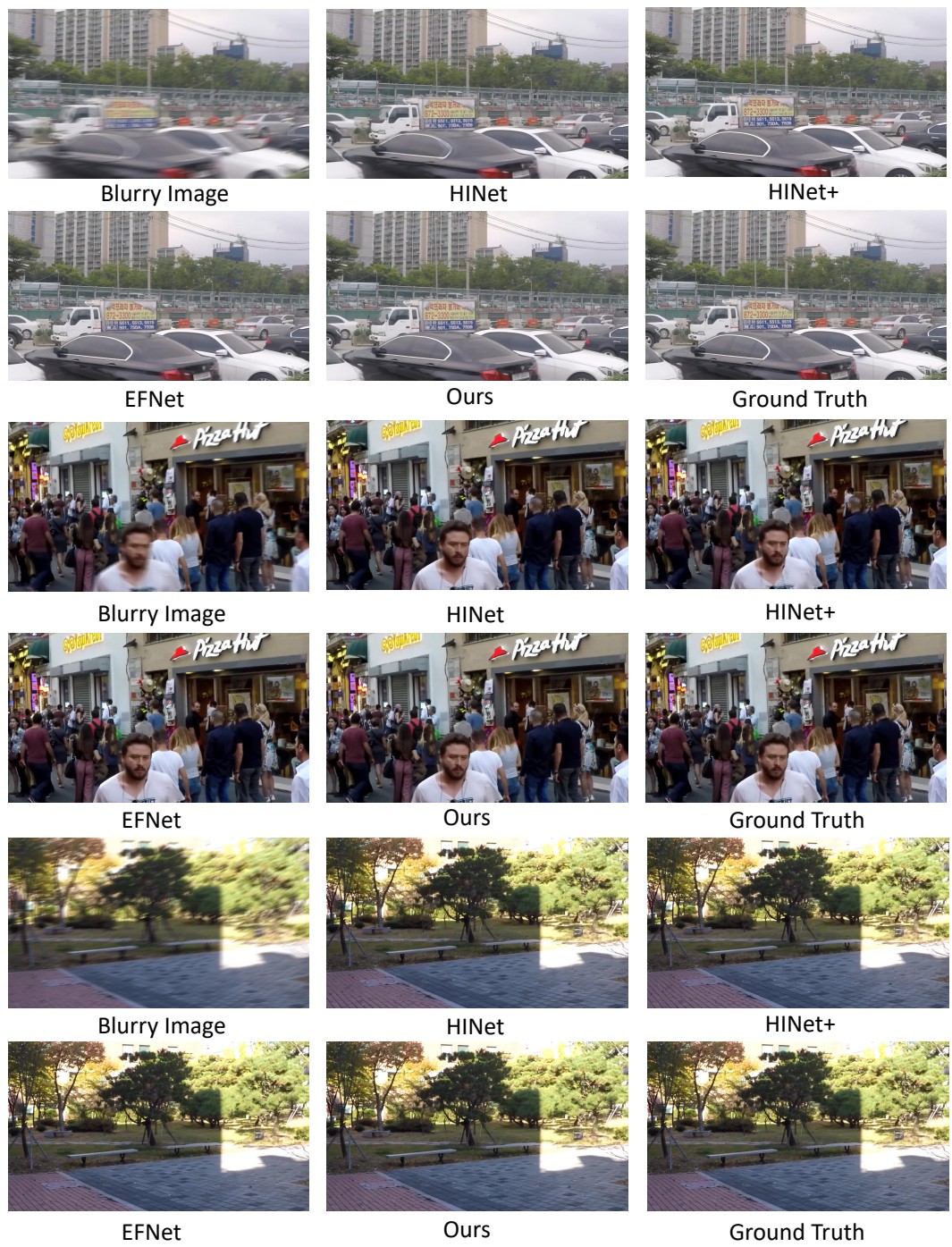

Figure S5: More Qualitative Results on Spk-GoPro Datasets.