# OpenReview forum: "Enhancing Motion Deblurring in High-Speed Scenes with Spike Streams"
_NeurIPS.cc/2023/Conference — NeurIPS 2023 poster_

### Official Review · Reviewer_kDLr · 2023-07-03

**Soundness:** 3 good
**Presentation:** 3 good
**Contribution:** 3 good
**Rating:** 5
**Confidence:** 4

**Summary:**

This paper proposed the first deblurring method that uses two types of inputs: RGB frames and spike streams. Two synthetic datasets are introduced. They are also used for training and evaluating the proposed method. This method outperforms other state-of-the-art methods on these synthetic datasets.

**Strengths:**

* This paper proposes the first spike-based motion deblurring model, which is based on transformers.
* The authors claim that they can reconstruct sub-frame sharp images at any timestamp, which is a great advantage.
* The datasets can be of research value to the community.
* The proposed method outperforms SOTA on those synthetic datasets.

**Weaknesses:**

* The datasets use synthetic blurry images and a synthetic spike camera simulator. The method is evaluated on a real spike stream only on a few images in the supplementary materials (which is not even mentioned in the main paper). Moreover, this brings up another issue: synchronizing a real spike camera with an RGB camera is not trivial. Some hybrid camera setup is shown in the supplementary materials, but it requires manually aligned spatial and temporal outputs. More thorough evaluation is needed.
* The main loss (13) is not clear. The first and the last terms seem the same. Why are they separated?
* The paper contains some typos: L205 (noisy), L76/L98 (Researches), L15 (vFurthermore)
* Intro: not true that deep learning models always find a clear image (L32)

**Questions:**

Addressing any of the above weaknesses would be great, especially about the real data.

**Limitations:**

Some limitations are addressed.

---

> ### Author Rebuttal · Authors · 2023-08-10
>
> Thank you for your helpful comments, summary of our paper, and affirmation of the performance. We would like to address your concerns and answer your questions here.
>
> ***1. More thorough evaluation in real-world scenarios is needed.***
>
> Thanks for your suggestion! Please refer to Response 1 in **To all reviewers**.
>
> ***2. The main loss (13) is not clear. The first and the last terms seem the same. Why are they separated?***
>
> Thanks for pointing it out. As presented in Sections 3.3 and 3.4.2, in the first term of the loss (Equation 13), $\hat{I}_t$ represents the final deblurred output from the deblurring branch, while in the third term, $\overline{I}_t$ represents the initial deblurred result generated by the deblurring branch. The separation of these terms serves a distinct purpose. We introduce $\overline{I}_t$ through the CAMMA module as a high-resolution image domain prior to guiding the spike branch to aid spike reconstruction. By performing the loss function on $\overline{I}_t$ and ${I}_t$, we aim to minimize the blurriness present in $\overline{I}_t$, thereby providing higher-quality image-domain priors to guide the spike branch's reconstruction. The loss function between $\overline{I}_t$ and ${I}_t$ constrains the final deblurring result. We will clarify it in the revised version.
>
> ***3. The paper contains some typos: L205 (noisy), L76/L98 (Researches), L15 (vFurthermore).***
>
> We greatly appreciate your meticulous review. We have revised these typos in the paper.
>
> ***4. Intro: not true that deep learning models always find a clear image (L32).***
>
> We acknowledge that our statement might not be precise. We have revised the statement to emphasize the potential of deep learning models when trained on large and diverse datasets, while also considering their limitations when trained with only a single modality as input.

---

> > ### Comment · Reviewer_kDLr · 2023-08-16
> > **After reading the rebuttal**
> >
> > Thank you for the rebuttal. My main concerns are mostly well addressed. I also read other reviews and responses to them. I believe they are also addressed well. Overall, this paper has no significant flaws, so I'm in favor of acceptance.

---

> > > ### Author Response · Authors · 2023-08-16
> > > **Thanks for your valuable time**
> > >
> > > Thank you sincerely for your insightful comments, valuable suggestions, and kind appreciation of our work. Thanks a lot for your valuable time!

---

### Official Review · Reviewer_9w9w · 2023-07-06

**Soundness:** 3 good
**Presentation:** 2 fair
**Contribution:** 2 fair
**Rating:** 4
**Confidence:** 5

**Summary:**

The paper attempts to remove motion blur from high-speed scenes making use of spike streams. Most deep learning-based deblurring algorithms predict sharp frames relying only on the input blurry frames and are not robust when the blurry artifact is severe. This work proposes to use spike streams that could be obtained from spike cameras along with the blurry input for motion deblurring. The authors propose a framework that integrates both modalities in a manner where information is shared bi-directionally. In addition, the work builds two synthetic datasets on top of the GoPro and X4K1000FPS datasets for training and evaluation. Experimental comparisons are presented with several baselines.

**Strengths:**

* The proposed problem formulation of motion deblurring using spike streams is quite new and a promising direction for further research
* The joint training of RGB and gray image reconstruction is an intuitive approach to enforce the proposed network to use both modalities
* The Spk-X4K1000FPS and Spk-GoPro datasets would be a useful contribution to the community if they become publicly available
* The qualitative results look good

**Weaknesses:**

* The motivation of the work is not convincingly justified

   There are several previous works [1,2] that use event information for motion deblurring. Compared to these works, it is not clear why the paper chooses spike information over events. The explanation in L52-53 that "...spike cameras record low-resolution texture information and ... this serves as a stronger guidance for deblurring task" is very vague and not convincing. What does low-resolution texture information have to do with motion deblurring? and why should it be a stronger guidance compared to high temporal resolution?

   The claim in L49-51 that, "Most event-based methods unidirectionally utilize information from the event domain to assist the image domain, without achieving the complementarity of information from both domains", is not true. There are several event-based works [1] that jointly use both the RGB and event information for motion deblurring. Hence, this claim should be toned down.

* The experimental results in the paper are weak and unfairly done

   The experimental comparison on X4K1000FPS in Table 1 is not very meaningful as it is clearly biased to the proposed method. The other baselines HiNet and NAFNet do not use extra data and hence, it is not surprising that they underperform. A more thorough and fair comparison should be done with previous event-based methods following their experiment protocol.

   The results for the GoPro dataset in Table 2 are simply copied (quoted) from previous papers. However, the authors follow a different experiment setting in their paper. Hence, how can the authors conclude that the performance gain is coming from the proposed approach and not necessarily from the different experimental protocols?

    The qualitative analysis both in the main paper and supplementary ignores to compare with the state-of-the-art approach, REFID [2]. Why is that?

* More ablations are needed to justify the different design choices in the proposed approach

   There has to be an ablation experiment that only uses the image stream to prove the benefit of incorporating the spike stream

   I want to see the ablation for using the claims in L188-193 (directly fusing the input blurred image with the spike branch). Removing the whole CSFI, CAMMA, and initial deblurring branch only results in a PSNR decrease of 0.2 dB. Hence, how worse would it be if we naively fuse the blurred input with the spike branch?

   What is the point of doing several experiments with different input representations? How does that lead to a better understanding of the proposed approach? I get that the spike branch leads to a good performance. However, all other modules seem to be very redundant with very minor contributions to network performance.

    It will be good to provide the visualization of the motion magnitude mask in L208, to verify if the CAMMA module is indeed doing what it is claimed to be doing.


* The writing of the paper could be improved

   The submission will greatly benefit from re-writing the methodology part by removing too many unnecessarily coined acronyms and by restructuring the explanation of the different modules in the proposed approach. Moreover, important details such as train-test set splits and results on real-world blurred images should be in the main paper instead of the supplementary.


References

[1]. Lei Sun, Christos Sakaridis, Jingyun Liang, Qi Jiang, Kailun Yang, Peng Sun, Yaozu Ye, Kaiwei Wang, and Luc Van Gool. Event-based fusion for motion deblurring with cross-modal attention, ECCV 2022

[2]. Lei Sun, Christos Sakaridis, Jingyun Liang, Peng Sun, Jiezhang Cao, Kai Zhang, Qi Jiang, Kaiwei Wang, and Luc Van Gool. Event-based frame interpolation with ad-hoc deblurring, CVPR 2023


**Questions:**

Please refer to the "Weaknesses" section and address the raised concerns carefully.

**Limitations:**

The authors adequately address the limitations of their work.

---

> ### Author Rebuttal · Authors · 2023-08-10
>
> ***1. What does low-resolution texture information have to do with motion deblurring? Why should it be a stronger guidance compared to high temporal resolution?***
>
> We would like to clarify that the low resolution of spike camera is due to current hardware limitation. Our emphasis is on the texture information within spike data (rather than resolution), which contributes positively to motion deblurring. To validate this assertion, We conducted a simple experiment. We use both the blurry RGB images in Spk-X4K1000FPS ($e=65$) and the grayscale images of the corresponding sharp RGB images as input to NAFNet for training. As shown in Tab.R4, we find that a grayscale image with rich texture can effectively guide the restoration of blurred RGB images.
>
> **Table.R4: Toy experiment using NAFNet.**
> |  Input    | PSNR ($e=65$) | SSIM ($e=65$) |
> |  ----  | ---- | ---- |
> | Blurry RGB  |29.06 |0.878 |
> | Blurry RGB + Gray Ground Truth |**39.84** |**0.999** |
>
> Thus, we assert that sharp texture information can effectively guide deblurring. In other words, the texture information within a grayscale image can remap color information in the blurred RGB image, even in cases of extreme blur ($e=65$). Our method explicitly reconstructs in the spike branch and strives to introduce sharper texture information into the deblurring branch to guide deblurring. Our CAMMA module aims at integrating image priors from the deblurring branch into the spike branch to enhance the spike reconstruction and subsequently provide improved guidance for the deblurring branch. Supplementary Tab.S1 demonstrates that the CAMMA enhances the quality of spike branch reconstruction. Tab.S2 illustrates that the CAMMA also improve the deblurring performance.
>
> Moreover, both temporal and texture information provides clues for reconstruction and deblurring. We apologize for the overly absolute description used in the paper. We will rectify this in the revised version.
>
> ***2. The claim in L49-51 that, "Most event-based methods unidirectionally utilize information from the event domain to assist the image domain" is not accurate.***
>
> Please refer to Response 2 in **To all reviewers**.
>
> ***3. A more thorough and fair comparison on X4K1000FPS should be done with previous event-based methods.***
>
> Please refer to Response 2 in **To all reviewers**.
>
> ***4. The authors follow a different experiment setting on GoPro dataset in their paper.***
>
> The data presented in Tab.2 is primarily cited from REFID[2]. We observed that REFID used different experiment settings in their experiments compared to EFNet[1]. This discrepancy could be attributed to REFID's utilization of recurrent units and multi-frame outputs, which may lead to higher GPU memory consumption and make it hard to use larger batchsize. In our case, we have conducted additional experiments on GoPro dataset using the same training settings as EFNet, as shown in Tab.R5.
>
> **Table.R5: Additional experiments of our method on GoPro dataset using the same training settings as EFNet.**
> |  Method    |Extra Data| PSNR  | SSIM  |
> |  ----  | ---- | ---- | ---- |
> | EFNet  |Event| 35.46 |0.972 |
> | REFID |Event|35.91 |0.973 |
> | Ours (old training settings) |Spike|36.12 |0.971 |
> | Ours (the same training settings as EFNet) |Spike|**36.83** |**0.976** |
>
> These results confirm the performance improvements achieved by our method even under EFNet's training parameter settings. We believe that this performance gain is attributed to higher learning rates and an increased number of total iterations (In our previous experiments, our network can converge in around 100k iterations, whereas EFNet requires approximately 300k iterations).
>
>
> ***5. The qualitative analysis ignores to compare with the state-of-the-art approach, REFID***
>
> We apologize for not including a comparison with REFID. Due to the lack of code before NeurIPS 2023 deadline and  suitable comparative images in the original REFID paper, we couldn't compare in the inital submission. But now with open-source code, we're addressing this gap by including a qualitative REFID comparison in the revised paper.
>
> ***6. The ablation experiment that only uses the image stream to prove the benefit of incorporating the spike stream.***
>
> We have conducted the requested ablation experiment. The results are provided in Tab.R6. This experiment validates that the introduction of spike data effectively improves performance.
>
> **Table.R6: Additional ablation experiments.**
> |  Method   | PSNR ($e=33/e=65$) | SSIM ($e=33/e=65$) |
> |  ----  | ---- | ---- |
> | Only uses the image stream  |32.45/28.25 |0.895/0.841 |
> | Our final SpkDeblurNet |**37.42/35.94** |**0.968/0.966** |
>
> ***7. The ablation for using the claims in L188-193 (directly fusing the input blurred image with the spike branch).***
>
> We have carried out the suggested experiments, involving the direct fusion of the input blurred image with the spike branch. The results are shown in Tab.R7. We observed that comparing to removing the CAMMA branch (eliminating information flow from image to spike branch), fusing the blurred image directly provides a performance gain, highlighting the importance of bidirectional information complementarity. With CAMMA, performance further improves, demonstrating its effectiveness in transferring image information to the spike branch efficiently.
>
> **Table.R7: Additional ablation experiments.**
> |  Method   | PSNR ($e=33/e=65$) | SSIM ($e=33/e=65$) |
> |  ----  | ---- | ---- |
> | Remove CAMMA branch  |37.20/35.64 |0.967/0.965 |
> | Directly fuse the input blurred image  |37.31/35.80 |0.967/0.965 |
> | Our final SpkDeblurNet |**37.42/35.94** |**0.968/0.966** |
>
> ***8. What is the point of doing several experiments with different input representations?***
>
> Please refer to Response 3 in To all reviewers.
>
> ***9. The visualization of the motion magnitude mask in L208.***
>
> Please refer to Response 4 in To all reviewers.
>
> ***10. About the writing.***
>
> We appreciate it and will carefully revise the writing.

---

> > ### Comment · Reviewer_9w9w · 2023-08-14
> >
> > I thank the authors for the rebuttal. I have read the rebuttal and other reviews. The following concerns remain unaddressed:
> >
> > 1. The authors do not directly answer the question of why low-resolution texture information (spike) should be stronger guidance compared to high temporal resolution (event) information as stated in Lines 52-53. The example presented in the rebuttal does not address this question since the problem setting is closer to a colorization task rather than a deblurring task. Given a motion blur is caused by a sudden camera or object motion, it makes sense to incorporate temporal (event) information for the deblurring task. How is texture information useful in this regard? Why should it be a stronger guidance than temporal information?
> >
> > 2. I understand why the authors could not compare with REFID during submission time. However, why did not the authors provide a qualitative comparison with REFID in the rebuttal PDF?
> >
> > 3.  I am not convinced about the effectiveness of the proposed CAMMA module. According to the authors' rebuttal, a simple fusion of spike information with the input blurred image already gives a strong performance and the addition of the CAMMA module seems to outperform this naive baseline by only 0.1dB. This also casts doubt on the importance of using information bi-directionally as a simple unidirectional baseline seems to perform on par with a baseline that uses bi-directional information.

---

> > > ### Author Response · Authors · 2023-08-15
> > > **Thank you for your patient reply**
> > >
> > > >1. The question of why low-resolution texture information (spike) should be stronger guidance compared to high temporal resolution (event) information as stated in Lines 52-53.
> > >
> > > (1) We apologize again for the overly absolute comparative term "stronger" in Line 53 and acknowledge that **both** temporal and texture information provide guidance for deblurring task, while the unique sampling mechanism makes spike cameras contain richer texture information than event cameras. We will revise the statement to emphasize that "**Both** the temporal and texture information serve as guidance for the deblurring task."
> > >
> > > (2) It's worth noting that the texture information contained in spike streams is **derived from temporal information [1]**. Specifically (as described in Sec.3.4.2), due to the proportional relationship between spike firing rate and light intensity, at a given position and time $t$, we can infer the pixel value by identifying the nearest preceding and succeeding spikes and calculating their time interval (also mentioned in Lines 52-53). Thus, spike cameras' texture information results from their dense temporal sampling, offering **more** valuable deblurring clues.
> > >
> > > (3) In the toy experiment, from the aspect of colorization, the network maps the blurred color information in the input RGB image back to the correct positions in the input sharp grayscale image, which in fact corresponds to deblurring process. So we consider the experiment as a deblurring task, which confirms that the extremely rich texture contained in the grayscale image can assist in deblurring.
> > > (Even if we regard it as a colorization task, due to the similarity between the latent texture information in spike stream and grayscale images (Fig.S4), our network indirectly achieves deblurring by coloring the potential sharp texture features based on the RGB information of the blurred image. In this perspective, the texture information in spike assists in deblurring.)
> > >
> > > To better clarify our argument, we conducted further experiments(Tab.R8) introducing both the blurred RGB image and the **LBP (Local binary patterns)** texture image of its ground-truth as inputs to NAFNet. This scenario is **distinct from colorization** and the results support the idea that diverse texture information can assist deblurring. That is to say, when possess texture information from a latent sharp image corresponding to the blurry image, the network can use it as clues to guide deblurring. The sharp edges in the texture may serve as anchors to guide deblurring, and notably, spikes inherently capture texture information. Thus it also makes sense to incorporate texture information for the deblurring task. We will further explore this in our future work.
> > >
> > > **Table.R8: More toy experiments using NAFNet.**
> > > |  Input    | PSNR ($e=65$) | SSIM ($e=65$) |
> > > |  ----  | ---- | ---- |
> > > | Blurry RGB  |29.06 |0.878 |
> > > | Blurry RGB + LBP of Ground Truth |35.90 |0.989 |
> > > | Blurry RGB + Gray Ground Truth |**39.84** |**0.999** |
> > >
> > > >2. Why didn't provide a qualitative comparison with REFID in the rebuttal PDF?
> > >
> > > We've evaluated and added the qualitative comparison with REFID in the revised version. However, due to the visual differences is minor, and space is constrained in the rebuttal PDF, we regret not including them. We promise to adding these visual results in the final version of the paper.
> > >
> > > >3. The effectiveness of the proposed CAMMA branch
> > >
> > > (1) The **CAMMA branch** we propose includes **two** key ideas: firstly, the introduction of information flow from the image domain to the spike domain, which leverages the rich texture characteristics of spike data, is distinct from event-based methods; secondly, the **CAMMA module** itself emphasizes sharp regions to further enhance performance. The combination of the two results in performance improvements from 0.2 to 0.4 dB across various input representations (Tab.3, Supp.Tab.S2). We gently believe that evaluating these ideas separately (Tab.R7) and considering each part has only a limited improvement would be not suitable, and we kindly hope that assessing these ideas collectively, recognizing the incremental gains from considering each component individually, could be a more appropriate approach. The mask visualization in Fig.R3 further validates the CAMMA module's efficacy. Tab.S1 also shows a performance gain of 0.4 to 3.6 dB for the spike reconstruction branch with the CAMMA branch across diverse input representations.
> > >
> > > (2) We observed that experiments directly fusing the blurred input tend to be smoother, while those using the CAMMA module retain more details in the visual results. We will include the relevant visual results in the future revised version.
> > >
> > > We plan to explore more concise spike-assisted deblurring approaches in the future.
> > >
> > > [1] Zhu L, Dong S, Huang T, et al. A retina-inspired sampling method for visual texture reconstruction[C]//2019 IEEE International Conference on Multimedia and Expo (ICME). IEEE, 2019: 1432-1437.

---

> > > > ### Author Response · Authors · 2023-08-15
> > > >
> > > > In our toy experiment, if we use the **Canny** edge map of the ground truth as auxiliary input, we obtain results with PSNR of 32.07 and SSIM of 0.942. This experiment also demonstrates that the edge texture information in the latent sharp image of the blurred image can serve as clues to guide the deblurring process.

---

> > > > > ### Author Response · Authors · 2023-08-19
> > > > > **Thank you for your valuable time and we hope our response will help your re-assessment of our work**
> > > > >
> > > > > Dear Reviewer 9w9w,
> > > > >
> > > > > Thank you sincerely for your constructive feedback. We notice that your concerns mainly lie in six parts in the previous review.
> > > > >
> > > > > 1. The first concern is about the relationship between the texture information and motion deblurring, for which we provided toy experiments with gray scale image/LBP texture map/Canny edge map of the ground truth as auxiliary input to show that texture information  can serve as clues to guide the deblurring process. We also apologize for the overly absolute comparative term and clarified the texture information in spike streams is derived from temporal information, and both are beneficial for deblurring.
> > > > > 2. The second concern is about some experiments settings, for which we have added the suggested comparison and abalation experiments.
> > > > > 3. The third concern is about the qualitative comparison with REFID, for which we have clarified the reason for not include them and promise to add these resuts in the revised version.
> > > > > 4. The fourth concern is about the effectiveness of the proposed CAMMA branch, for which we have provided related ablation experiments in Tab.3, Supp.Tab.S1&S2 and Tab.R7. We also highlighted the the incremental gains in the CAMMA branch.
> > > > > 5. The fifth concern is about the point of experiments with different input representations, for which we clarified that they presented CAMMA's potential to enhance performance across diverse inputs.
> > > > > 6. The sixth concern is about the visualization of the motion magnitude mask, for which we have provided the masks and showcase the effectiveness of CAMMA.
> > > > >
> > > > > We would like to express our sincere gratitude again for your valuable comments and nice suggestions. During the rebuttal phase, we have tried our best to eliminate the confusion and improve our paper. Since the discussion period is approaching its end, we would be glad to hear from you about whether our responses have addressed your concerns. We are sincerely looking forward to your further reply.

---

### Official Review · Reviewer_wVmq · 2023-07-06

**Soundness:** 2 fair
**Presentation:** 2 fair
**Contribution:** 2 fair
**Rating:** 5
**Confidence:** 5

**Summary:**

This paper proposes a motion deblurring method that integrates RGB images and binary spike streams. In detail, it has a content-aware motion magnitude attention module and a transposed cross-modal attention fusion module. Experiments demonstrate state-of-the-art performance on deblurring datasets.

**Strengths:**

1, the first spike-based motion deblurring model.

2, the two large-scale synthesized datasets for spike-based motion deblurring

3, state-of-the-art results.

**Weaknesses:**

1, The Transposed Cross-Attention Fusion (TCAF) is close to the multi-Dconv head transposed attention (MDTA) in Restormer.

2, The TCAF module is similar to EICA in EFNet.

3, The spike camera is similar to event camera, and the method based on these two methods are also similar except the format of event/spike stream. I suspect that event-based methods also works with spike camera. Because the paper mainly forcuses on the algorithm design, and the proposed SpkDeblurNet makes no huge differences with the existing EFNet, from this aspect the novelty of the paper is limited.

4, In the abstract, "preserving rich spatial details" and "are low-resolution image signals" are conflicting and confusing.

5, Typo in abstract.

**Questions:**

See Weaknesses

**Limitations:**

See Weaknesses

---

> ### Author Rebuttal · Authors · 2023-08-10
>
> Thank you for your summary, and we appreciate you for pointing out the strengths of our paper. My clarification and answers for the weaknesses you summarized are as follows.
>
> ***1. The Transposed Cross-Attention Fusion (TCAF) is close to the multi-Dconv head transposed attention (MDTA) in Restormer, and the TCAF module is similar to EICA in EFNet.***
>
> Thanks for pointing it out. As described in Section 3.4.3, unlike the original usage of MDTA solely for self-attention computation, in TCAF, we have adapted and modified the MDTA module proposed in Restormer to address two specific challenges in our multi-modal fusion: 1) The lightweight nature of the spike branch with fewer channels compared to the deblurring branch makes conventional spatial attention computation not directly applicable. 2) The computational cost of traditional attention mechanisms grows quadratically with input size, which is impractical for image restoration tasks.
>
> Furthermore, as an effective module, the MDTA module has been drawn upon by multiple studies[1,2]. In contrast to EFNet's EICA module, our TCAF differs in that the two modalities within our module possess differing channel counts, with the spike branch having fewer channels. We also employ depth-wise convolutions to enhance local context before computing the attention map, aiming to improve generalization.
>
> ***2. The spike camera is similar to the event camera, and the method based on these two methods are also similar except for the format of the event/spike stream. I suspect that event-based methods also work with spike camera.***
>
> Please refer to Response 2 in **To all reviewers**.
>
> ***3. In the abstract, "preserving rich spatial details" and "are low-resolution image signals" are conflicting and confusing.***
>
> We would like to clarify that the phrase "preserving rich spatial details" refers to the spike cameras' ability, **in contrast to event cameras** that primarily capture temporal information, to capture per-pixel spatial information alongside temporal data. On the other hand, "are low-resolution image signals" reflects the current state of spike cameras, where the sensor array is limited to $250\times 400$ photosensitive units, resulting in low-resolution signals **compared to high-resolution RGB images**. We will revise the wording in the revised version to mitigate any potential misunderstandings.
>
> ***4. Typo in the abstract.***
>
> Thank you for your careful observation. We have revised these typos in the paper.
>
> [1] Sun L, Sakaridis C, Liang J, et al. Event-based fusion for motion deblurring with cross-modal attention[C]//European Conference on Computer Vision. Cham: Springer Nature Switzerland, 2022: 412-428.
>
> [2] Song J, Mou C, Wang S, et al. Optimization-Inspired Cross-Attention Transformer for Compressive Sensing[C]//Proceedings of the IEEE/CVF Conference on Computer Vision and Pattern Recognition. 2023: 6174-6184.
>
> [3] Sun L, Sakaridis C, Liang J, et al. Event-Based Frame Interpolation with Ad-hoc Deblurring[C]//Proceedings of the IEEE/CVF Conference on Computer Vision and Pattern Recognition. 2023: 18043-18052.

---

> > ### Author Response · Authors · 2023-08-16
> > **Thanks for your valuable time and hope our responses helpful for your re-assessment of our work**
> >
> > Thank you for the thorough feedback and constructive suggestions. We would like to kindly inquire whether our previous response clarified your concerns and if there are any additional comments. We are glad to cooperate and provide answers to assist in the review process. Thank you very much for your time!

---

> > > ### Author Response · Authors · 2023-08-19
> > > **Restatement of our responses that we hope can help for your re-assessment**
> > >
> > > Dear Reviewer wVmq,
> > >
> > > Thank you sincerely for your detailed feedbacks. We notice that in your initial review, your concerns lie in three parts.
> > >
> > > 1. The first concern is the difference of our TCAF module from the previous works, for which we have clarified the differences from existing modules and our motivation for the adaptation.
> > > 2. The second concern is whether event-based methods can work with spike camera, for which we clarified the
> > > intrinsic distinctions between the cameras and methods, and we provided a comparison to show our approach's superiority with spike data.
> > > 3. The third concern is about the confusing phrase, for which we clarifed the exact meaning we want to express and promise to revise the wording in the revised version.
> > >
> > > Based on these facts and positive feedback from other reviewers, we sincerely inquire that if our previous response could settle and answer your concerns and kindly hope you could re-consider your initial rating. If you still have any further comments or questions, please let us know and we are glad to address your further concerns.

---

> > ### Comment · Reviewer_wVmq · 2023-08-20
> >
> > Thanks for your reply. My concerns are largely addressed, so I update my score to borderline accept. Although the reply 1 is still arguable, this paper may be useful for the community. Please update the comparison and new results to the paper.

---

### Official Review · Reviewer_DRAW · 2023-07-06

**Soundness:** 3 good
**Presentation:** 3 good
**Contribution:** 2 fair
**Rating:** 5
**Confidence:** 4

**Summary:**

The paper proposes a novel approach that integrates the two modalities from two branches, leveraging spike streams as auxiliary visual cues for guiding deblurring in high-speed motion scenes, introducing a content-aware motion magnitude attention module and transposed
 cross-modal attention fusion module.

**Strengths:**

1. Authors propose the first spike-based motion deblurring model equipped with content-aware motion magnitude attention and a cross-modal transposed attention fusion module.

2. Extensive experiments have demonstrated the effectiveness of the proposed model

3. Overall, the paper is well-written and technically sound.


**Weaknesses:**

1. Although the authors conducted the validation on two synthetic datasets that the proposed method is effective, the performance on real scenarios is still unknown. Especially, it is hard to simultaneously obtain aligned pairs of spike streams and RGB images. So, have authors ever considered constructing a camera system to truly capture the spike data and blur images to better demonstrate the proposed setup is feasible and has practical value ?

2. Considering the extra cost of spike cameras, is it really worth using such an expensive device for deblurring rather than using a high-speed global shutter camera directly capture relatively sharp images? Authors should provide more motivations for using spike cameras from the perspective of reality.


3. What about the generalization ability of the proposed method? Authors should present related experiments. Moreover, the computational complexity and inference time are also supposed to be compared.

4. Some minor issues: typos (e.g. line 15)

**Questions:**

See weaknesses.

**Limitations:**

See weaknesses.

I encourage the authors to actively and rigorously prepare the rebuttal and I will raise the rating if my concerns are well-addressed.

---

> ### Author Rebuttal · Authors · 2023-08-10
>
> Thank you for your positive and constructive feedback. We are encouraged that you find our method effective. We would like to address your concerns and answer your questions here.
>
> ***1. Have authors ever considered constructing a camera system to truly capture the spike data and blur images to better demonstrate the proposed setup is feasible and has practical value ?***
>
> Thanks for your suggestion. Please refer to Response 1 in **To all reviewers**.
>
> ***2. Authors should provide more motivations for using spike cameras from the perspective of reality.***
>
> We would like to clarify that the spike camera is built using the same CMOS sensors as traditional cameras, and it leverages consumer-grade integrated circuits through regular semiconductor manufacturing processes, making it a cost-effective solution [1]. In contrast, conventional high-speed cameras, such as Phantom cameras, necessitate specialized sensors and shutters that are considerably expensive. Specifically, the spike camera employs an innovative temporal domain sampling method with continuous photon capture by photosensitive units, as opposed to synchronous exposure with a fixed exposure time. When accumulated intensity surpasses a predefined threshold, a spike is generated. These spikes, generated by each photosensitive unit, are spatially organized to form a spike stream array. This sampling technique enables achieving exceptionally high sampling rates on conventional CMOS sensors. Considering the low cost of the spike camera and its advantages of high speed and high dynamic imaging and recording rich spatial texture information, we believe that the spike camera holds promising prospects for various applications.
>
> ***3. What about the generalization ability of the proposed method? Moreover, the computational complexity and inference time are also supposed to be compared.}***
>
> We showcase visual results in Supplementary Fig. S2 by applying our model trained on the Spk-X4K1000FPS dataset with a window size of 33 to real-world sequences. These results demonstrate the generalization ability of our method to real data.
> Additionally, we conduct experiments to test models trained on Spk-X4K1000FPS ($e=33$) on data with $e=65$, as well as cross-dataset testing by applying models trained on Spk-X4K1000FPS to the GoPro dataset. The visualized results are provided in the PDF file of the global response. All these results demonstrate the robust generalization of our method across diverse blurring conditions and scenes.
>
> We have added comparisons regarding computational complexity and inference time in Tab.R3 (note that the settings for e=33 and e=65 are only applicable to the Spk-X4K1000FPS dataset). Our approach achieves a favorable balance between complexity and performance, as evident from our results.
>
> **Table.R3: Comparisons between different methods regarding computational complexity and inference time.**
> |  Method    | MACs | Params | Inference Time | PSNR (on GoPro) |
> |  ----  | ---- | ---- |---- |---- |
> | HINet  |170.49G |88.67M |20.2ms |33.69|
> | NAFNet  |63.06G|67.79M |27.8ms |33.69|
> | EFNet  |107.93G |7.73M |14.9ms |35.46|
> | REFID  |4.36T |88.81M |781.2ms |35.91|
> | Ours ($e=65$)  |53.25G |12.93M |140.6ms |N/A|
> | Ours ($e=33$)  |53.18G |12.92M |107.9ms |N/A|
> | Ours ($e=56$)  |53.23G |12.93M |130.1ms |36.12|
>
>
>
> ***4. Some minor issues: typos (e.g. line 15)}***
>
> We appreciate your keen attention to detail. The mentioned typos have been rectified in the revised version of the paper.
>
> [1] Huang T, Zheng Y, Yu Z, et al. 1000× faster camera and machine vision with ordinary devices[J]. Engineering, 2022.

---

> > ### Author Response · Authors · 2023-08-16
> > **Thanks for your feedbacks and we are wondering whether we have addressed your concerns**
> >
> > Thank you for the detailed feedbacks and constructive suggestions. We sincerely hope our posted response can help to address your concerns on our paper and serve as a reference for your re-assessment of our work. If you have any further comments and questions, please let us know and we are glad to write a follow-up response. Thank you very much!

---

> > ### Comment · Reviewer_DRAW · 2023-08-18
> >
> > Thanks for the response.
> >
> > After reading the authors' rebuttals and other reviewers' comments, I will hold my ratting.

---

> > > ### Author Response · Authors · 2023-08-19
> > > **Thanks for your precious time and we would like to see if there is any further concern and comment**
> > >
> > > Dear Reviewer DRAW,
> > >
> > > Thank you sincerely for your insightful feedbacks. We notice that in your initial review, your concerns lie in three parts.
> > >
> > > 1. The first concern is about the **real-world camera system**, for which we have provided detailed information about the prototype of our hybrid camera system in Supplementary Sec.D.1 and Fig.S1. Deblurred results can be found in Fig.S2 and Fig.R4.
> > > 2. The second concern is the **motivations** for using spike cameras, for which we explain that the spike camera is built with consumer-grade CMOS sensors and integrated circuits through regular semiconductor manufacturing processes, making it a cost-effective solution for various applications.
> > > 3. The third concern is about the
> > > **generalization ability and the computational complexity**, for which we provided experiments of the favorable generalization ability to real data, diverse blurring conditions and datasets, and we added comparisons about computational complexity.
> > >
> > > Given these facts and positive feedbacks from other reviewers, we would like to kindly ask again if our previous response clarifies your concerns and if this could potentially serve as a basis for improving your initial rating. Also, if you have any further questions or comments, please let us know, and we are glad to give further responses and clarification.

---

### Official Review · Reviewer_Rugk · 2023-07-07

**Soundness:** 3 good
**Presentation:** 3 good
**Contribution:** 3 good
**Rating:** 6
**Confidence:** 4

**Summary:**

This paper introduces a novel approach that combines traditional cameras and spike cameras to address motion blur in high-speed scenes. By leveraging spike streams as auxiliary visual cues, the proposed spike-based motion deblurring model effectively extracts relevant information from blurry images using content-aware motion magnitude attention and transposed cross-modal attention fusion modules.

**Strengths:**

The paper proposes to novel technique to deblur a blurry scene using a traditional camera frame along with spike camera frames. The proposed method is able to leverage the high SNR information from the traditional camera frame and the motion information from the spike camera frames to reconstruct the final image with high quality.

**Weaknesses:**

The authors have missed a major branch of related works using Quanta Image Sensors (QIS). QIS are a very similar family of image sensors that operate at high speed and produce single-bit frames, and has a lot of previous works on deblurring and denoising. For eg.,  [1] Ma, S., Gupta, S., Ulku, A.C., Bruschini, C., Charbon, E. and Gupta, M., 2020. Quanta burst photography. ACM Transactions on Graphics (TOG), 39(4), pp.79-1. [2] Chi, Y., Gnanasambandam, A., Koltun, V. and Chan, S.H., 2020. Dynamic low-light imaging with quanta image sensors. In Computer Vision–ECCV 2020: 16th European Conference, Glasgow, UK, August 23–28, 2020, Proceedings, Part XXI 16 (pp. 122-138). Springer International Publishing. [3] Chandramouli, P., Burri, S., Bruschini, C., Charbon, E. and Kolb, A., 2019, May. A bit too much? High speed imaging from sparse photon counts. In 2019 IEEE International Conference on Computational Photography (ICCP) (pp. 1-9). IEEE.

In fact, "[4] Liu, Y., Gutierrez-Barragan, F., Ingle, A., Gupta, M. and Velten, A., 2022. Single-photon camera guided extreme dynamic range imaging. In Proceedings of the IEEE/CVF Winter Conference on Applications of Computer Vision (pp. 1575-1585). " uses a very similar idea of combining two imaging modalities, but for a different task - HDR imaging.

EDIT:
With the promised change, my major concern will be addressed.

**Questions:**

The related works and the comparison section should ideally have a detailed comparison with the QIS based methods.

**Limitations:**

Yes

---

> ### Author Rebuttal · Authors · 2023-08-10
>
> ***1. The authors have missed a major branch of related works using Quanta Image Sensors (QIS). The related works and the comparison section should ideally have a detailed comparison with the QIS-based methods.}***
>
> Thanks for recommending these works. We appreciate the recognition of Quanta Image Sensors' (QIS) ability to capture high-speed motion and perform well across various tasks. In the revised paper, we have thoroughly reviewed and incorporated the following references[1-4] into our introduction, related work, and comparative experiments. We will incorporate the discussion on these works in our final version, where the one additional page allows us to extend the current Sections with more content and illustrations.
>
> Furthermore, we would like to gently emphasize that while both the spike camera and QIS are capable of capturing high-speed scenes, there exist significant differences between the two technologies. QIS primarily relies on the single-photon avalanche diode (SPAD) detector technology, whereas the spike camera is constructed from CMOS sensors similar to traditional cameras and utilizes regular semiconductor manufacturing processes, resulting in a more cost-effective solution. Exploiting the fact that the time sensitivity of CMOS photosensitive devices widely used today has reached tens of nanoseconds, the spike camera achieves exceptional temporal resolution by mimicking the sampling mechanism of the primate fovea.
>
> Additionally, we would like to clarify that the works of [1-3] focus on reconstruction tasks specific to QIS and involve single-modal data only. Moreover, while both our method and the QIS-assisted HDR imaging method[4] incorporate two modalities, this does not imply a similarity between our work and [4]. Firstly, multi-sensor modality fusion is a common task in the field of autonomous driving. Secondly, [4] simply concatenates the features of two modalities during skip connections for fusion. In contrast, our method takes a bidirectional complementary approach to leverage information from both modalities. We introduce the initial deblurred output from the image branch as a high-resolution image domain prior to guiding the spike branch to aid spike reconstruction, and then we introduce the refined spike features into the image branch to guide deblurring in the image domain. Additionally, we propose the TCAF module for the effective fusion of the two branches.
>
> Since our method and [4] are two different tasks and cannot be directly compared, to validate the effectiveness of our proposed framework, we adapted the network of [4] for our spike-assisted deblurring task under their experiment protocol. Experimental results in Tab.R0 indicate that our framework achieves superior performance over [4] and can better utilize the complementarity between different modalities to improve performance.
>
> **Table.R2: Comparisons between different networks on Spk-X4K1000FPS dataset.**
> |  Method   | PSNR ($e=33/e=65$) | SSIM ($e=33/e=65$) |
> |  ----  | ---- | ---- |
> |  CMOS-SPC [4]  |33.65/31.59 |0.926/0.912 |
> | SpkDeblurNet (Ours) |**37.42/35.94** |**0.968/0.966** |
>
> [1] Ma, S., Gupta, S., Ulku, A.C., Bruschini, C., Charbon, E. and Gupta, M., 2020. Quanta burst photography. ACM Transactions on Graphics (TOG), 39(4), pp.79-1.
>
> [2] Chi, Y., Gnanasambandam, A., Koltun, V. and Chan, S.H., 2020. Dynamic low-light imaging with quanta image sensors. In Computer Vision–ECCV 2020: 16th European Conference, Glasgow, UK, August 23–28, 2020, Proceedings, Part XXI 16 (pp. 122-138). Springer International Publishing.
>
> [3] Chandramouli, P., Burri, S., Bruschini, C., Charbon, E. and Kolb, A., 2019, May. A bit too much? High speed imaging from sparse photon counts. In 2019 IEEE International Conference on Computational Photography (ICCP) (pp. 1-9). IEEE.
>
> [4] Liu, Y., Gutierrez-Barragan, F., Ingle, A., Gupta, M. and Velten, A., 2022. Single-photon camera guided extreme dynamic range imaging. In Proceedings of the IEEE/CVF Winter Conference on Applications of Computer Vision (pp. 1575-1585).

---

> > ### Comment · Reviewer_Rugk · 2023-08-14
> >
> > Thank you for your reply. With the promised change, my major concern will be addressed.

---

> > > ### Author Response · Authors · 2023-08-16
> > > **Thanks for your valuable time**
> > >
> > > Thank you very much for the thorough review and for increasing the score. Your time and input mean a lot to us.

---

### Author Rebuttal · Authors · 2023-08-10

We appreciate all reviewers for their helpful feedbacks. We are encouraged that they found our paper "well-written and technically sound" [DRAW], our method "novel" [Rugk] and "intuitive" [9w9w], and the proposed problem "new" [9w9w] and "is a promising direction for further research". We are delighted that they acknowledge that our method outperforms state-of-the-art methods [wVmq,kDLr] and the qualitative results are good [9w9w]. We are glad that they recognize our dataset [wVmq,9w9w,kDLr] and evaluate it as "a useful contribution"[9w9w] with "research value"[kDLr]. We will incorporate all feedback.

In this general response, we would like to address some cruical concenrns.

***1. More evaluation in real-world scenarios.***
> **[Reviewer DRAW]** Have authors ever considered constructing a camera system to truly capture the spike data and blur images to better demonstrate the proposed setup is feasible and has practical value?

Yes, actually we have provided relevant information about the prototype of our hybrid camera system in Supplementary Sec.D.1 and Fig.S1. We used this system to capture pairs of blurred images and spike sequences from real-world scenes and applied the proposed method to obtain deblurred results, as shown in Supplementary Fig.S2.

> **[Reviewer kDLr]** More thorough evaluation in real-world scenario is needed.

We achieve temporal synchronization by simultaneously triggering the capture programs of both cameras through a script. We use a feature point matching algorithm to achieve spatial registration. We are currently in the process of improving the hybrid camera system prototype and extending our evaluation to include a broader range of real-world scenarios. We have added additional results in real-world scenes in Fig.R4 in the PDF in the global response.

***2. More comparative experiments that applying event-based methods to spike camera.***
> **[Reviewer wVmq]** The spike camera is similar to event camera, and the method based on these two methods are also similar except the format of event/spike stream. I suspect that event-based methods also works with spike camera.

> **[Reviewer 9w9w]** A more thorough and fair comparison on X4K1000FPS should be done with previous event-based methods.

While both camera types are neuromorphic cameras, event cameras utilize differential sampling, while spike cameras employ integral sampling. This fundamental difference leads to intrinsic distinctions. Both capture rich temporal information, yet spike cameras uniquely excel in capturing detailed spatial-temporal textures. This trait motivates our incorporation of image-domain priors into spike domains for explicit reconstruction. These refined spike features enhance the image branch for guiding deblurring. The mutual complementation of information between modalities enhances performance. In contrast, event camera-based methods often involve one-way information flow from the event branch to the image branch, highlighting a significant difference from our approach.

Despite these disparities, both approaches fall under the multi-modal paradigm. Hence, we conducted supplemental experiments using EFNet and REFID networks for spike-assisted deblurring. For EFNet, we employed input representations similar to its SCER approach, excluding the EMGC module due to its event-related nature. Regarding REFID, we utilized similar spike voxel representations. Tab.R1 results demonstrate our approach's superiority with spike data, yielding enhanced outcomes.


**Table.R1: Comparisons between different networks on Spk-X4K1000FPS dataset.**
|  Method   | PSNR ($e=33/e=65$) | SSIM ($e=33/e=65$) |
|  ----  | ---- | ---- |
| EFNet  |36.36/33.53 |0.960/0.937 |
| REFID  |36.30/33.47 |0.962/0.945 |
| SpkDeblurNet (Ours)  |**37.42/35.94** |**0.968/0.966** |

>**[Reviewer 9w9w]** The claim in L49-51 that, "Most event-based methods unidirectionally utilize information ... without achieving the complementarity of information" is not accurate.

Our phrase "achieving the complementarity of information from both domains" signifies bidirectional information flow between modal branches, enhancing each other's tasks. EFNet, in contrast, unidirectionally incorporates event data into the image branch for RGB deblurring, without reciprocating by including image data into the event branch. Our approach facilitates mutual bidirectional information exchange between branches, fostering mutual assistance. We will include a clearer elaboration in the revised version.

***3.[Reviewer 9w9w] What is the point of doing several experiments with different input representations?***

Apart from validating enhanced deblurring performance with diverse input representations via the spike branch, our motivation for exploring different input representations also involves evaluating the effectiveness of our CAMMA module. Supplementary Tab.S1 shows CAMMA's enhancement of spike branch reconstruction quality across various inputs. Tab.S2 shows CAMMA's contribution to improved overall deblurring performance. These experiments present CAMMA's potential to enhance performance across diverse inputs.


***4. [Reviewer 9w9w] The visualization of the motion magnitude mask in L208.***

We've added visualizations of the motion magnitude mask in our global response PDF. In Fig.R3, we display the hard thresholding mask and the content-aware mask for both spike stream and CSFI input representations. The hard thresholding mask roughly masks high motion magnitude regions, while the content-aware mask offers refined masking. Notably, the CSFI representation yields a more pronounced content-aware mask due to its reduced information content compared to the spike stream, requiring greater reliance on image-domain prior knowledge. Supplementary Tab.S1 and Tab.S2 also demonstrate the significant improvement brought about by the introduction of CAMMA module when utilizing the CSFI representation.

---

### Author Response · Authors · 2023-08-13
**Look Forward to Feedbacks**

Dear Reviewers,

Thank you sincerely for your review. We would greatly appreciate it if you could inform us of any remaining questions or concerns that you may have, so that we can address them promptly prior to the deadline.  Alternatively, if you feel that your initial concerns are addressed, we would appreciate updating your evaluation to reflect that. Thank you!

---

### Decision · Program_Chairs · 2023-09-21

**Decision:**

Accept (poster)

**Comment:**

The paper has positive final ratings except for a single reviewer who articulated certain remaining concerns and gave a score of borderline reject. The AC read the paper, all the reviews, the authors' responses, and discussion.

After taking everything into consideration, the AC finds the paper to be novel in proposing the use of spike cameras (along with RGB ones) to help with motion de-blurring. The inference approach is also sufficiently interesting/novel to clear the bar.

The AC acknowledges the BR reviewer's main remaining complaints, but does not find them to be a bar for acceptance.

(1) The AC thinks the question of whether there's enough "motivation/intuition" for why spike cameras would be better than event cameras is moot --- intuition is never going to be precise, and the experimental results show their method is better than results reported with event cameras (reviewer complains about not comparing with a CVPR 2023 paper in the main paper and only giving quantitative comparisons in the rebuttal --- the AC thinks that's reasonable on part of the authors, the quantitative results are better, it's contemporary work anyway, and in any event, the novelty and impact of using a new kind of camera for deblurring cues is novel).

Having said that, authors should tone down the assertion that spike data is intrinsically more informative than event data for deblurring.

(2) The fact that one component of the method showed limited impact in ablations is also not a bar --- at least the ablation is there, and so the reader has all the information they need.

Therefore, the AC is happy to recommend acceptance.